# AUTO-SCALING CONTINUOUS MEMORY FOR GUI AGENT

## ABSTRACT

We study how to endow GUI agents with scalable memory that help generalize across unfamiliar interfaces and long-horizon tasks. Prior GUI agents compress past trajectories into text tokens, which balloons context length and misses decisive visual cues (*e.g.,* exact widget size and position). We propose a continuous memory that encodes each GUI trajectory into a fixed-length sequence of continuous embeddings using the VLM itself as an encoder; these embeddings are plugged directly into the backbone's input layer, sharply reducing context cost while preserving fine-grained visual information. As memory size and retrieval depth increase, performance improves monotonically, unlike text memories that degrade with long prompts. To grow memory at low cost, we introduce an auto-scaling data flywheel that (i) discovers new environments via search, (ii) synthesizes tasks with an open-source VLM, (iii) rolls out trajectories with the agent, and (iv) verifies success with the same VLM. Using this pipeline, we collect 10k trajectories for about $500 and fine-tune only the memory encoder (LoRA on a Q-Former, 1.2% parameters) with 1,500 samples. On real-world GUI benchmarks, our memory-augmented agent consistently improves success rates under long horizons and distribution shifts. Notably, Qwen-2.5-VL-7B + continuous memory achieves performance comparable to state-of-the-art closed-source models (*e.g.,* GPT-4o, Claude-4). Our data and code will be publicly released.

## 1 INTRODUCTION

Recent advances in visual grounding and training techniques for vision–language models (VLMs) (McKinzie et al., 2024; Wu et al., 2025) have catalyzed rapid progress in Graphical User Interface (GUI) agents (Zhu et al., 2025; Team, 2025a). With appropriately designed frameworks and action spaces, these agents can operate across websites, desktop software, and mobile apps to solve multi-step planning tasks such as web search and online shopping. However, real-world tasks are often more unpredictable and long-horizon. They demand robust generalization to unfamiliar visual layouts, iconography, and unseen functionalities that require new knowledge. Existing GUI agents generally perform not well under such distribution shifts (Li et al., 2024; Lu et al., 2025a), incurring repeated retries or execution failures on complex plans. By contrast, humans exhibit strong robustness across diverse GUIs. Through drawing on accumulated episodic experiences and encountered interface states from human memory, we can easily learn to use new webs and apps, and accomplish novel tasks (Tulving, 2002). Moreover, human memory is continually refreshed. We ingest information from varied sources to retain potentially useful cues for future scenarios (O'Reilly et al., 2014).

As a promising path toward better generalization, recent work equips GUI agents with memories built from collected GUI trajectories (Fang et al., 2025; Zhang et al., 2025). Each trajectory is a sequence of screenshots and executed actions. Without compression, a single trajectory can span thousands, sometimes hundreds of thousands of tokens, inflating inference cost and introducing irrelevant detail. To control cost, prior systems typically compress trajectories into text-only discrete tokens. However, text-based representations cannot faithfully capture crucial visual cues in GUI environments (*e.g.,* the precise size and position of clickable elements), which are often decisive for reliable execution.

To overcome these limitations, we encode GUI trajectories into compact, transferable **continuous embeddings**. Concretely, following the paradigm that uses the VLM itself as the encoder (Wu et al., 2025), we build a continuous memory collection in which each trajectory is compressed into a

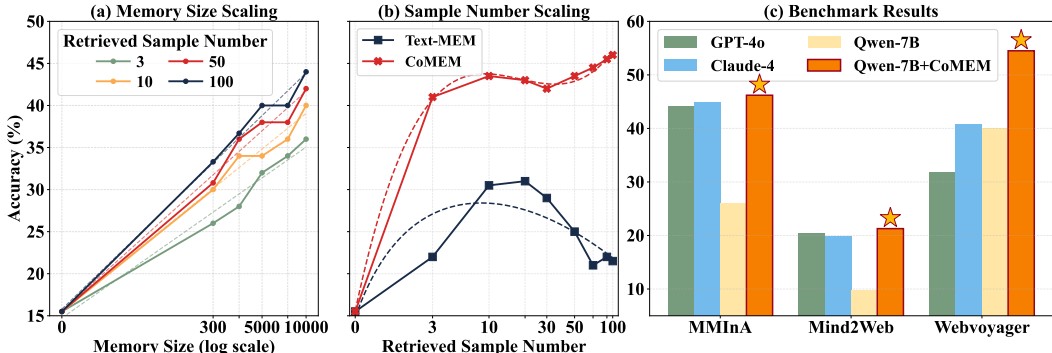

Figure 1: Experimental results to verify the effectiveness of our continuous memory in GUI agent setting. (a) Performance scaling law with the memory size; (b) Performance scaling law with retrieved sample number; (c) CoMEM outperforms closed-source state-of-the-art models across benchmarks.

fixed-length sequence of 8 embeddings. These condensed memories sharply reduce context length while can preserve fine-grained visual cues. At inference, retrieved memory embeddings are injected directly into the VLM's embedding layer to plug in new knowledge. This model-friendly integration alleviates the context-length bottleneck and supports scaling both the total memory size and the number of in-context retrieved items. As shown in Figure 1, performance with continuous memory improves monotonically as we scale memory size and retrieval depth, exhibiting a clear scaling trend, whereas text-based memories degrade beyond roughly ten retrieved items due to ballooning sequence length, increased attention overhead, and accumulated semantic noise during inference.

To capitalize on this scaling law, we seek to scale continuous memory by collecting far more GUI trajectories. Yet high-quality traces are typically human-annotated, which are costly and hard to scale up. We therefore propose an **auto-scaling data flywheel** that autonomously discovers new environments, creates tasks, rolls out trajectories, and checks quality. Concretely, we first crawl candidate websites or apps through the search engine, and then leverage an open-source VLM to synthesize the task queries for the new environment. Next, we utilize the agent model to perform plan-and-action to solve the synthetic task on the environment, and continue to use the VLM to check if the task goal is achieved. In practice, we found that open-source VLMs' planning and verification capabilities are sufficient to maintain diversity and quality at scale. Thus, the whole data flywheel only needs a search engine, a well-performed open-source VLM, and an agent model, which enables continual and low-cost auto-scaling of a highly informative continuous memory for GUI agents.

Based on the data flywheel, we spend $553 to collect 15,145 GUI trajectories for building the continuous memory, and utilize a very efficient fine-tuning paradigm that only optimizes the LoRA (Hu et al., 2021) and Q-Former(Li et al., 2023) layer in the memory encoder using totally 1,500 training samples and 1.2% parameters. Extensive experiments demonstrate that our approach leads to consistent improvements on real-world GUI benchmarks and also guarantees a relatively lower inference cost, including long-horizon and distribution shifts scenarios. As shown in Figure 1, Qwen-2.5-VL-7B (Yang et al., 2024) equipped with our memory method can achieve comparable performance with state-of-the-art closed-source models (*e.g.,* GPT-4o (OpenAI, 2024a) and Claude-4(Anthropic, 2024)), and even significantly outperform them on the Webvoyager dataset.

In summary, the contributions of this paper are as follows:

- We introduce a continuous memory for GUI agents that encodes trajectories as compact, plug-and-play embeddings for frozen VLM backbones, yielding monotonic gains as memory size and retrieved items scale.

- We build an auto-scaling data flywheel that discovers environments, synthesizes tasks, rolls out trajectories, and performs quality control to expand a diverse and high-quality memory without human annotation.

- Extensive experiments on diverse GUI planning tasks show that memory-augmented agents perform better than state-of-the-art models.

## 2 RELATED WORK

**GUI Agents.** A wave of benchmarks has standardized evaluation for GUI agents: Mind2Web (Deng et al., 2023) targets generalist web control across many real sites; WebArena (Zhou et al., 2023) and VisualWebArena (Koh et al., 2024) add realistic, execution-based evaluation for text-only and multimodal agents; OSWorld (Xie et al., 2024) scales to real operating systems and open-ended desktop workflows and underscores agents' current limitations in grounding and operational knowledge; and GUIOdyssey (Lu et al., 2025a) aims for cross-app mobile GUI navigation. Across benchmarks, modern GUI agents improve via better grounding (Li et al., 2025; Wu et al., 2024), larger and more diverse training (Qin et al., 2025), and RL-style finetuning (Luo et al., 2025; Lu et al., 2025b). Specifically, WebGPT (Nakano et al., 2022) fine-tunes an LLM to browse and cite sources in a text-only environment; WebShop agents (Yao et al., 2022) that interleave reasoning with text actions and have been applied to web tasks. Beyond DOM-centric pipelines, a parallel line focuses on *screenshot-based* agents. SeeAct (Zheng et al., 2024) frames GPT-4V as a generalist web agent operating from screenshots, with a tool interface and online evaluation on live websites. WebSight (Bhathal & Gupta, 2025) likewise adopts a vision-first pipeline that eliminates DOM reliance, pairing a fine-tuned VLM with modular planning/verification and episodic memory. UI-TARS (Qin et al., 2025) reports strong end-to-end agent performance via large-scale GUI pretraining and iterative data collection. Our setting follows this screenshot-only, tool-calling paradigm and extends it with continuous memory to enable compact, transferable knowledge reuse.

**Memory for LLMs and Agents.** LLM/VLM agents remain constrained by finite context windows and weak long-term retention, motivating external and learned memory mechanisms. Early retrieval-based approaches such as RAG (Lewis et al., 2020) and REALM (Guu et al., 2020) equips models with non-parametric textual memory, but long concatenated contexts raise inference cost and can distract planning (Arslan et al., 2024). Recent works move beyond token concatenation toward *structured* and *continuous* memories. VoCo-LLaMA (Ye et al., 2025) and MA-LMM (He et al., 2024a) compress visual content into compact embeddings. CoMEM (Wu et al., 2025) advances this line for VLMs by learning continuous memory that is portable across models and avoids the pitfalls of token concatenation. In the agent literature, Agent Workflow Memory (Wang et al., 2025) induces reusable workflows from experience and selectively retrieves them to guide future decisions, improving long-horizon tasks both offline and online. Memp (Fang et al., 2025) targets procedural memory, distilling past trajectories into step-level instructions and higher-level scripts with explicit build–retrieve–update mechanisms for lifelong use. (Zhang et al., 2025) proposes trajectory-level retrieval over unified, multimodal agent traces, positioning trajectory banks as first-class memory for agent policy reuse and analysis. Together, these directions suggest that dense, model-agnostic memories can provide stable, low-overhead knowledge that transfers across tasks and backbones.

**LLM for Data Annotation.** A growing body of work replaces costly human supervision with LLM-driven synthesis and grading. Self-Instruct (Wang et al., 2023) and its evolutionary variants (Zeng et al., 2024; Xu et al., 2024) bootstrap large, scalable instruction datasets directly from pretrained LLMs' own generations. In GUI domains, ZeroGUI (Yang et al., 2025) automates online learning by having a VLM generate tasks from the current screen, roll out policies, and verify success, thereby expanding training data at near-zero human cost. SEAgent (Sun et al., 2025) is an agentic self-evolving framework enabling computer use agents to autonomously evolve through interactions with unfamiliar software, therefore empowers mastering novel software environments via experiential learning. Together, these works provide a practical recipe for continual data growth to fuel agent learning and, in our case, to expand the memory at scale.

## 3 PRELIMINARY

**Problem Statement.** Following the *vision-based* setting in SeeAct (Zheng et al., 2024) and WebSight (Bhathal & Gupta, 2025), we consider a GUI interaction environment that exposes a page rendering engine and a set of action interfaces. Given the environment and the natural language task instruction (*e.g.,* "*Find the address for the nearest Armageddon Shop*"), the agent perceives the world solely through pixels and selects the next action from the following set:

$$\mathcal{A} = \{\text{CLICK, TYPE, SCROLL, WAIT, STOP}\}. \tag{1}$$

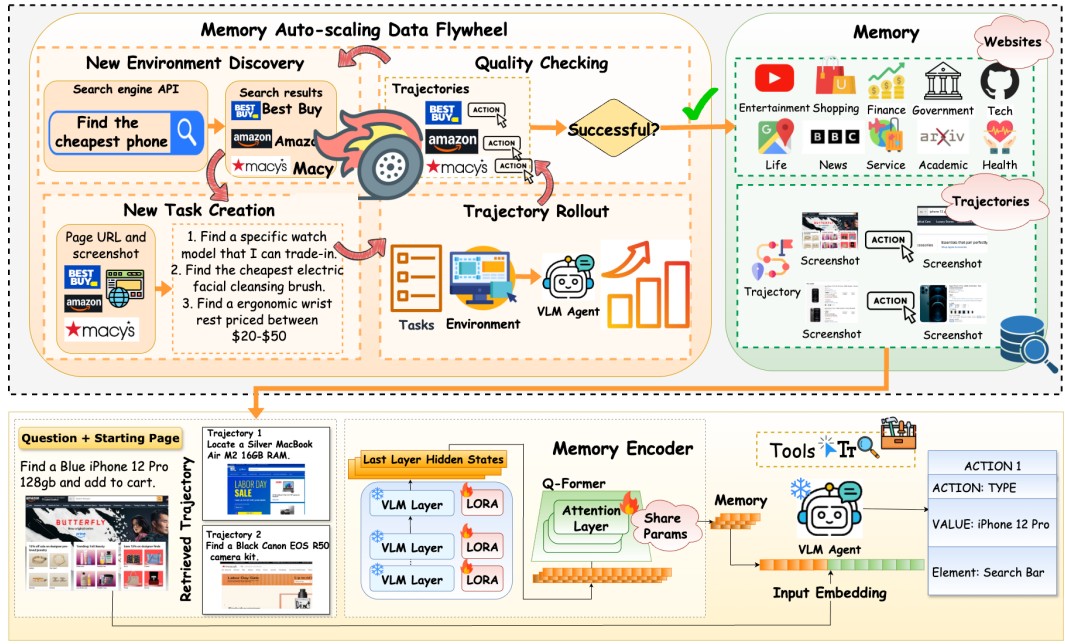

Figure 2: Overview of our memory-augmented VLM agent framework. We devise (1) a four-phase memory auto-scaling data flywheel; (2) a memory storing diverse environments and successful trajectories; and (3) a VLM-based memory encoder that converts retrieved trajectories from the memory into compact embeddings to guide the VLM during inference.

At time $t$, the agent receives an observation $o_t = \langle I_t \rangle$ from the current state, where $I_t$ is a screenshot of the GUI environment. Then, the agent outputs a structured action call $a_t \in \mathcal{A}$ to operate the GUI. The environment transition executes $a_t$ on the interface and returns the next screenshot $I_{t+1}$, yielding a trajectory $\tau = (o_t, a_t)_{t=1}^{T}$. We assume a horizon cap $T_{max}$. Following existing work (He et al., 2024b), a task is *successful* if the last $s$ page states satisfy the goal according to *LLM-as-Judge evaluation*. We report standard success rate on public web-agent benchmarks.

**Our Target.** In this paper, we aim to equip the GUI agent with a continuous memory that consists of compressed embeddings of useful trajectories from different environments and tasks. For each task, we first retrieve the top-$k$ relevant items from the memory, and then concatenate their continuous embeddings into the context. At each step, the agent model produces the next action conditioned on both the current observation and memory items, and the memory-augmented policy is:

$$a_t \sim \pi_\theta(a \mid o_t, m_t), \tag{2}$$

where $m_t$ is the concatenated continuous vectors of the retrieved memory items.

## 4 APPROACH

According to the promising improvement and scaling law shown in Figure 1, we aim to equip the GUI agent with a scalable continuous memory containing useful GUI trajectories. We first introduce our devised data flywheel for automatically scaling the memory, and then present the integration of our continuous memory and the agent model. We show the overview of our approach in Figure 2.

### 4.1 MEMORY AUTO-SCALING DATA FLYWHEEL

Large-scale, diverse, high-quality GUI trajectories are key for our continuous memory to strengthen GUI agents' capabilities. To enable scaling, we introduce an autonomous data flywheel that grows the memory without any human intervention. For a start, let $\mathcal{Q}$ be the task/query pool, $\mathcal{E}$ the environment pool (web/apps), and $\mathcal{T}$ the trajectory pool. We initialize $\mathcal{Q}_0$ using the queries from a seed dataset, *i.e.,*

Mind2Web training set (Deng et al., 2023; Zheng et al., 2024), and set $\mathcal{E}_0$ and $\mathcal{T}_0$ as empty set $\varnothing$. At each iteration, we update $(\mathcal{Q}_t, \mathcal{E}_t, \mathcal{T}_t) \rightarrow (\mathcal{Q}_{t+1}, \mathcal{E}_{t+1}, \mathcal{T}_{t+1})$ through four phases that continuously discover new environments, synthesize tasks, roll out trajectories, and conduct quality evaluation.

**Phase-1: New Environment Discovery.**  Inspired by humans who often seek new apps and webs through online search, we utilize an online search engine to discover diverse new environments. We first sample a few queries from $\mathcal{Q}_t$, and use SerpAPI[1] to search for the top 20 related websites. Then, we simply test the stability and accessibility of the websites, and filter out the low-quality ones. Next, we deduplicate the websites with $\mathcal{E}_t$, and obtain the newly discovered environment set $\mathcal{E}^*$.

**Phase-2: New Task Creation.**  For each new environment $e \in \mathcal{E}^*$, we use a VLM to automatically synthesize new task queries for it. Given the environment screenshot, the VLM first produces a detailed text description of the site. Using this description, along with the screenshot, the VLM then generates a set of candidate task queries $\mathcal{Q}_e^*$ that are potentially solvable within the environment.

**Phase-3: Trajectory Rollout.**  Next, we employ an agent model (*i.e.,* Qwen2.5-VL-32B (Yang et al., 2024)) to perform autonomous rollouts. Given each query $q \in \mathcal{Q}_e^*$, we follow the setup described in Section 3 to make the agent model interact with environment $e$. We collect all the actions and observations to compose the trajectory $\tau_q^* = (o_t, a_t)_{t=1}^T$.

**Phase-4: Quality Checking.**  Upon completion of each task, we evaluate the resulting trajectory using a dedicated judging model. The evaluation process involves feeding the task query $q$ and the full trajectory $\tau_q^*$ into the judge VLM, which determines whether the task was successfully completed. For high-quality evaluation, we use SEAgent-1.0-7B (Sun et al., 2025), a fine-tuned model specifically trained for GUI agent assessment. Finally, we collect all the positive trajectories, and their task queries and environments, to update the trajectory pool $\mathcal{T}_t$, task pool $\mathcal{Q}_t$, and environment pool $\mathcal{E}_t$.

**Outcome.**  This *discover* → *generate* → *rollout* → *verify* closed loop provides a simple but flexible auto-scaling strategy to continuously expand $\mathcal{Q}$, $\mathcal{E}$, and $\mathcal{T}$. In practice, by setting simple diversity and quality filters, we can further prevent redundancy and ensure robustness. In experiments, we spend about $553 to collect 15,145 trajectories spanning from 6,676 environments. As shown in Table 1, our trajectory dataset is a more diverse large-scale dataset than existing opensource ones.  Our data fly-

Table 1: Comparison across different GUI data resource. # Env. and # Samples denote the number of collected environments and samples in the datasets, respectively. Step-wise and Fully Auto Anno. denote whether we use fine-grained step-wise annotation and fully automatic annotation strategy, respectively.

| Dataset | # Env. | # Samples | Step-wise Anno. | Fully Auto Anno. |
|---|---|---|---|---|
| WebArena | 4 | 812 | ✓ | ✗ |
| VisualWebArena | 3 | 910 | ✓ | ✗ |
| WebShop | 1 | 12,087 | ✗ | ✗ |
| OmniACT | 30 | 9,802 | ✗ | ✗ |
| OSWorld | 10 | 369 | ✗ | ✗ |
| ScreenAgent | 39 | 723 | ✗ | ✗ |
| Mind2Web | 137 | 2,350 | ✓ | ✗ |
| Webvoyager | 15 | 643 | ✗ | ✓ |
| **Ours** | **6,676** | **15,145** | ✓ | ✓ |

wheel supports full step-wise and automatic annotations, enabling low-cost scale-up with minimal manual effort. More details can be found in Appendix B.

## 4.2 INTEGRATING CONTINUOUS MEMORY TO GUI AGENT

We first introduce the design of the memory encoder, then the fine-tuning strategy and the retrieval mechanism. Leveraging the trajectories collected by our data flywheel, we integrate a continuous memory into the GUI agent to enable effective knowledge transfer across tasks and environments. We first describe the memory encoder, then the retrieval mechanism, and finally the fine-tuning strategy.

---

[1]https://serpapi.com/

**Continuous Memory Encoder.** Following CoMEM (Wu et al., 2025), we employ a Q-Former (Li et al., 2023) to compress each retrieved multimodal trajectory into a small set of continuous memory embeddings. These embeddings are injected into the agent in a plug-and-play manner by prepending them to the model's input embeddings, allowing the agent to attend to contextual memory without architectural changes or full retraining. Thanks to the high representational capacity of continuous embeddings, long trajectories that often exceed 15,000 tokens[2] can be compressed to as few as 8 vectors, making large external knowledge and history feasible within limited context budgets.

**Memory Retrieval.** At inference time, given the current observation $o_t$, we perform embedding-based multimodal retrieval to fetch the most relevant prior experiences. Concretely, a CLIP encoder (Radford et al., 2021) maps screenshots and associated actions/queries from each stored trajectory to a set of embeddings, which are then pooled into a single multimodal key. We index all keys with FAISS (Douze et al., 2024) and retrieve the top-$k$ nearest neighbors. The corresponding trajectories are converted by the memory encoder into continuous embeddings and prepended to the agent's input, providing exemplar-driven guidance without inflating the token prompt.

**Efficient Fine-tuning for Memory Encoder.** To adapt the memory encoder to the agent with minimal cost, we fine-tune it using LoRA (Hu et al., 2021) (rank 16) shared across all Q-Former (Li et al., 2023) layers, updating only 1.2% of parameters. For data efficiency, we train on 1,500 high-quality trajectories drawn from open-source and synthetic sources. Each step in a trajectory forms a training instance augmented with its top-3 retrieved memories. This parameter- and data-efficient setup completes in 20 hours on a single NVIDIA H100 while yielding strong generalization.

## 5 EXPERIMENTS

### 5.1 EXPERIMENTAL SETUP

**ReAct-style and Tool-calling Mechanism.** Our main actor model adopts a vision-language architecture aligned with the tool-calling and ReAct paradigms. To execute actions, the model outputs structured reasoning followed by a tool invocation from a predefined toolset consisting of both GUI tools (e.g., CLICK, TYPE, SCROLL PAGE, WAIT, STOP) and Analysis tools (e.g., Page Content Analyzer, Change Page). For location grounding—such as selecting a pixel for clicking or typing, we augment the screenshot using a SOM-based labeling method, assigning clear identifiers to each UI element. The model must describe the target item and specify its label. If it fails to produce a valid label, we fall back to UI-TARS-1.5-7B (Qin et al., 2025) as a backup grounding module to ensure robust interaction. This hybrid approach enables both accurate decision-making and reliable UI manipulation in complex web environments.

**Evaluation Settings.** Experiments are conducted on three challenging multimodal web-agent benchmarks, MMInA (Zhang et al., 2024), Multimodal-Mind2Web (Deng et al., 2023; Zheng et al., 2024) and Webvoyager (He et al., 2024b), covering diverse real-world web-using scenarios.

(i) **MMInA** (Zhang et al., 2024) is designed to evaluate GUI agents on real-world websites. It contains 1,050 tasks across domains such as shopping and travel, requiring agents to perform multimodal grounding and long-horizon planning. We use the Wikipedia and Shopping domains and test on all available samples within these two categories.

(i) **Multimodal-Mind2Web** (Deng et al., 2023; Zheng et al., 2024) is a large-scale benchmark containing over 2,000 open-ended tasks from 137 real-world websites spanning 31 domains. For our evaluation, we select the first 100 tasks from the test-domain and test-website subsets, which include domains and websites not seen during training, thereby evaluating the out-of-distribution (OOD) generalizability of the agent. We skip tasks where the associated website is no longer accessible at evaluation time.

(i) **WebVoyager** (He et al., 2024b) contains real-world tasks from 15 websites and challenges agents to ground and reason in highly dynamic and multimodal web environments. We follow WebSight (Bhathal & Gupta, 2025), to use a subset of achievable tasks for evaluation.

---

[2]Each trajectory typically contains about 10 action–screenshot pairs; each pair costs 1,500 tokens in common VLMs (Zheng et al., 2024).

We adopt the LLM-as-Judge paradigm (He et al., 2024b) to evaluate task completion. For MMInA, we provide the language model with both the model's final answer and the ground truth answer, and ask it to determine whether the response is correct. For Multimodal-Mind2Web and WebVoyager, we supply the task description along with a sequence of trajectory screenshots to a VLM, which judges whether the task has been successfully completed. We report Task Accuracy as the evaluation metric.

**Baseline Methods.** To assess the effectiveness of our proposed memory-augmented framework, we compare the performance of the actor model across the following four settings:

(i) **Closed Source Base Model:** including GPT-4o (OpenAI, 2024b), Gemini-Pro-Vision (Google, 2023), and Claude-4(Anthropic, 2024).

(ii) **Open Source Base Model:** including GLM 4.1V-9B (Team, 2025b), Qwen2.5-VL-7B (Wang et al., 2024), and Qwen2.5-VL-32B (Yang et al., 2024). These two settings serve as baselines without task-specific adaptation or external memory.

(iii) **Specialized Fine-Tuned Model:** including UI-TARS-1.5 (Qin et al., 2025), CogAgent (Hong et al., 2023), and WebSight (Bhathal & Gupta, 2025), which are fine-tuned specifically for GUI tasks. This setting provides a comparison with task-specific adaptation.

(iv) **Open Source Model + Memory:** where we augment the base model with external memories in two different forms. *Text-based Memory* includes only unimodal (text) external experience trajectories in the form of tokenized textual prompts, serving as a baseline to assess the impact of text-based memory. *CoMEM* compresses and stores multimodal (text and screenshots) external experience trajectories in a continuous embedding space, evaluating the benefits of rich, multimodal memory representations for complex agent tasks.

All results reported in Table 2 are based on our own reproduction. The environment setup, task sampling, and evaluation protocols are consistent to ensure fair comparison across models.

## 5.2 MAIN RESULTS

Table 2: Performance comparison of GUI Agents on MMInA, Mind2Web, and WebVoyager evaluation of task accuracy. Bold indicates the best performance, and underline denotes the second-best. Results from closed-source base models are for reference only and excluded from ranking.

| Model | MMInA | | Mind2Web | | | | WebVoyager | Avg. |
|---|---|---|---|---|---|---|---|---|
| | Wiki | Shop | Shop | Travel | Info | Service | Overall | |
| **Closed Source** | | | | | | | | |
| GPT-4o | 51.3% | 37.0% | 15.4% | 14.3% | 22.6% | 29.4% | 31.8% | 27.8% |
| Gemini-Pro-Vision | 52.3% | 41.6% | 12.5% | 25.0% | 20.8% | 22.8% | 47.7% | 30.4% |
| Claude-4 | 50.0% | 40.0% | 10.5% | 22.2% | 19.8% | 26.7% | 40.9% | 28.8% |
| **Open Source** | | | | | | | | |
| Qwen2-VL-7B | 7.8% | 0.0% | 0.0% | 2.2% | 8.3% | 14.0% | 31.8% | 8.8% |
| Qwen2.5-VL-7B | 36.7% | 15.5% | 2.6% | 9.5% | 9.6% | 17.3% | 40.0% | 14.4% |
| GLM 4.1V-9B | 34.7% | 20.3% | 13.3% | 11.1% | 13.6% | **33.3%** | 40.0% | 23.0% |
| Qwen2.5-VL-32B | 43.3% | 37.6% | 8.0% | 12.2% | 7.6% | 13.0% | 40.9% | 21.6% |
| **Specialized Finetuned** | | | | | | | | |
| UI-TARS-1.5 | 36.4% | 1.0% | 0.0% | 14.3% | 5.6% | 6.5% | 34.8% | 13.2% |
| CogAgent | 20.5% | 7.0% | 10.7% | **20.0%** | 12.4% | 20.6% | - | 15.3% |
| Websight | 12.0% | 9.5% | 8.3% | 6.7% | 13.3% | 17.6% | 47.7% | 15.8% |
| **Memory-augmented** | | | | | | | | |
| UI-TARS-1.5-7B | | | | | | | | |
| + Text-based Memory | 16.0% | 1.0% | 0.0% | 11.0% | 3.6% | 8.6% | 34.0% | 10.0% |
| + CoMEM | 41.3% | 17.9% | 14.3% | 18.2% | 23.3% | 18.9% | 38.0% | 23.8% |
| Qwen2.5-VL-7B | | | | | | | | |
| + Text-based Memory | 34.2% | 31.4% | 7.1% | 17.8% | 12.7% | 16.6% | 44.0% | 22.2% |
| + CoMEM | **47.4%** | **45.0%** | **22.2%** | 18.8% | **26.5%** | 17.7% | **54.5%** | **31.7%** |

**Result Analysis.** As shown in Table 2, general base models perform poorly on web GUI tasks, with Qwen2.5-VL-7B achieving only 26.11% task accuracy in MMInA and 9.78% in Mind2Web. These models lack grounding in interactive environments and struggle with the structured reasoning required for web interfaces. Specialized finetune models such as CogAgent and WebSight perform better in some domains (e.g., CogAgent achieves 15.94% in Mind2Web tasks), but remain inconsistent and lack robustness across tasks, suggesting limited generalization beyond their training data.

Augmenting the base model with external memories significantly improves performance, as it provides additional context, past experience, and task-relevant knowledge. For example, adding external Text-based Memory raises the performance of Qwen2.5-VL-7B to 32.78% in MMInA, 44.0% in Webvoyager. Our proposed CoMEM further boosts performance by augmenting rich multimodal trajectories into compact embeddings. With CoMEM, Qwen2.5-VL-7B achieves 46.20% in MMInA, 21.28% in Mind2Web and 54.5% in Webvoyager, which is the best performance across all settings. This demonstrates that continuous memory effectively encodes long, multimodal experience into very few tokens, enabling efficient and scalable memory integration.

We also observe that UI-TARS-1.5-7B performs poorly across benchmarks, primarily due to its lack of exposure to interactive web navigation tasks during training. However, when augmented with CoMEM, UI-TARS-1.5 achieves a substantial performance improvement—from just 6.6% to 23.8% overall—demonstrating that our memory system effectively compensates for planning limitations by providing structured, multimodal task experiences that guide decision-making.

**Scaling Law.** To characterize the scaling behavior of our continuous memory, we refer to the empirical tendency, and leverage a rather simple log-linear function to model the relationship between model accuracy and the memory size $M$:

$$\text{Acc}(m) = a + b \log m \tag{3}$$

and estimate $\langle a, b \rangle$ by ordinary least squares separately for each fixed number of memories samples $m \in \{3, 10, 50, 100\}$. As seen Figure 1(a), the log-linear function can well describe the relationship and indicates the constantly increasing gains with respect to the scaling of the memory size. Besides, when comparing the curves with different retrieved samples, it is obvious that more samples lead to steeper improvement when increasing the memory size.

For the effect of the number of retrieved memory samples $K$, we we fit a cubic polynomial in $\log k$ to accuracy on MMInA Shopping tasks via ordinary least squares. As shown in Figure 1(b), CoMEM exhibits a sustained upward trend, whereas text-based memories peak and then deteriorate beyond roughly ten retrieved items, likely due to ballooning sequence length and accumulated noise.

## 5.3 FURTHER ANALYSIS

Table 3: Out-of-domain-GUI Environment Evaluation. AMS and SR denote Action Matching Score and Success Rate respectively. We evaluate Qwen-VL on GUI-Oddsey and UI-TARS on OSWorld.

| Model | GUI-Oddsey (AMS) | | OSWorld (SR) | | | |
|---|---|---|---|---|---|---|
| | High Level | Low Level | Office | Daily | Professional | Overall |
| Baseline | 22.38% | **45.58%** | 24.70% | 25.60% | 60.20% | 26.40% |
| + Text-based Memory | 24.42% | 37.35% | 23.10% | 23.10% | 57.10% | 24.70% |
| + CoMEM | **27.41%** | 44.90% | **25.13%** | **28.21%** | **60.87%** | **26.73%** |

**Out-of-domain GUI Environment.** To evaluate the generalization capability of our continuous memory mechanism, we conduct experiments in out-of-domain (OOD) GUI environments. Specifically, we utilize the memory constructed in web-based environments and a memory encoder fine-tuned on the web data. Evaluation is performed on two GUI benchmarks: GUI-Odyssey (Lu et al., 2025a) and OSWorld (Xie et al., 2024). OSWorld focuses on real-world desktop operating systems and open-ended workflows, while GUI-Odyssey targets cross-application navigation on mobile GUIs.

Following the original evaluation protocols, we report the Action Matching Score (AMS) for GUI-Odyssey and the Task Success Rate (SR) for OSWorld. The results reveal two key findings: (1) the Text-based Memory exhibits a negative impact on performance in OOD settings. Since it relies on

copying explicit actions or formats from prompts, it often introduces noise when applied to unfamiliar environments. (2) In contrast, the continuous memory demonstrates strong generalization. It encodes abstract, high-level knowledge and actionable rules into compact embeddings, which transfer effectively across different GUI domains—including unseen operating systems and applications.

**Inference Latency Study.** To assess the impact of memory augmentation on agent efficiency, we measure the average time consumption per trajectory across two tasks in MMInA (Zhang et al., 2024): *Wikipedia* browsing and *Shopping*, as shown in Table 4. Importantly, our method does not introduce additional time latency during inference. In fact, for the Wikipedia task, both memory-augmented variants, Text-based Memory and CoMEM, demonstrate even shorter completion times compared to the baseline. This suggests that access to prior experience allows the agent to make more informed decisions and follow more efficient trajectories.

Table 4: Inference speed and performance comparison across different methods. We report the average inference time (in minutes) per trajectory and accuracy (Acc.) in two domains.

| Method | Wikipedia | | Shopping | |
|---|---|---|---|---|
| | Time | Acc. | Time | Acc. |
| Qwen2.5-VL | 2.33 | 72.4 | 1.57 | 68.9 |
| + Text-based Memory | 1.50 | 77.1 | 2.12 | 73.5 |
| + CoMEM | 1.58 | **81.3** | 2.13 | **76.8** |

Although the task completion time for the Shopping task is slightly higher with memory integration, the difference is marginal (within 0.5 minutes) and does not indicate any substantial overhead. Overall, the results confirm that our memory mechanism maintains time efficiency while providing the added benefit of improved task performance and generalization.

**Training Efficiency** To evaluate the efficiency of our training process, we conduct experiments using varying amounts of high-quality training trajectories for aligning the compressed memory embeddings with the VLM's representation space. As shown in Table 5, our method achieves strong performance with limited training data. Notably, with only 1500 high-quality trajectories, the model reaches peak performance on both the Wikipedia (47.40%) and Shop-ping (45.00%) tasks. This indicates that our memory encoder can effectively distill and align semantic

Table 5: Task success rate comparison of using different training data scale.

| Data Size | Wikipedia | Shopping |
|---|---|---|
| 500 | 39.30% | 33.30% |
| 1000 | 43.83% | 39.00% |
| 1500 | **47.40%** | **45.00%** |
| 2000 | 45.00% | 42.60% |

knowledge into a compact embedding space with minimal data. Interestingly, increasing the training data to 2000 trajectories does not yield further gains, suggesting that our model is already well-aligned and sample-efficient. These results highlight the practicality of our approach in real-world scenarios where labeled trajectories are expensive or scarce.

## 6 CONCLUSION

We presented a memory-augmented framework for GUI agents that tackles long-horizon generalization by encoding prior trajectories as compact continuous embeddings and scaling this memory through an autonomous data flywheel. The proposed memory greatly compresses multimodal trajectories while preserving critical visual cues, enabling practical retrieval augmentation at inference without ballooning context length. We empirically verified that the performance improves monotonically as we scale memory size and retrieval depth. To sustain growth, we devised an auto-scaling data flywheel to collect diverse and high-quality experiences, consisting of environment discovery, task synthesis, trajectory rollout, and quality checking. Using only a search engine, an open-source VLM, and an agent, our data flywheel yielded 15,145 trajectories at low cost ($553). Based on the collected memory data, we adopted a lightweight adaptation regime (LoRA + Q-Former; 1.2% parameters on 1.5k samples) to equip a 7B open model with our memory. Experimental results have shown that the memory-augmented 7B model can attain accuracy competitive with leading closed-source systems.

Overall, our results indicate that continuous memory and autonomous experience acquisition form a robust foundation for scalable, generalizable GUI agents. Future work includes adaptive retrieval policies conditioned on uncertainty, tighter integration with RL for credit assignment over memories, extension to mobile and desktop automation at scale, and privacy-aware on-device memorization.

## LIMITATION AND ETHICS STATEMENT

This work outlines a practical path to scaling GUI agents by shifting knowledge from long token histories to compact, learnable embeddings and fueling them with autonomously harvested trajectories, but several technical limits remain. Retrieval can drift under extreme UI shifts (novel layouts, widgets, interaction patterns), and screenshot-only inputs underrepresent non-visual states; expanding retrieval to execution traces or UI graphs may help. At memory scale, freshness, deduplication, and provenance are hard to govern, and larger banks stress latency and GPU memory; age-/domain-aware eviction, clustering-based dedup, hierarchical or sharded FAISS indexes, and on-device caches are natural next steps. Finally, benchmark coverage lags real-world non-stationarity and evolving websites, complicating exact reproducibility; releasing crawl seeds, environment allowlists, and versioned testbeds can improve transparency.

Our data flywheel—environment discovery, task synthesis, rollout, and judging—also introduces reliability and bias risks. VLM judges can admit failures or reject valid successes, and self-reinforcing loops may overfit popular layouts or "easy" sites; ensemble judging, disagreement sampling with targeted spot checks, and domain-balanced crawling/retrieval mitigate this. The pipeline is vulnerable to data poisoning (malicious pages, adversarial screenshots) and prompt-injection-style UI content; source filtering, allowlists, per-memory safety scans, and retrieval-time trust scoring are required. Because auto-collected pages may embed copyrighted assets or restrictive licenses, public releases should prioritize features/embeddings plus URLs and honor site-specific terms.

Ethical considerations center on safety, privacy, and environmental cost. Autonomous rollouts must never operate on real accounts or payment flows; sandboxing, dry-run modes, click whitelists, ToS/robots checks, and fail-safes are mandatory. Screenshots can contain PII or sensitive content; apply consented collection only, automated redaction, minimization, strict access controls, and documented retention/erasure policies. Although our training is lightweight, large-scale crawl/embedding/indexing consumes energy; tracking usage, using efficient encoders, and sharing public indexes can reduce footprint. Addressing these limitations—alongside the modeling advances—is essential for safe, privacy-preserving, and reproducible deployment of memory-augmented GUI agents.

## REPRODUCIBILITY STATEMENT

To ensure the reproducibility of our work, we have uploaded our complete source code as a zip file in the supplementary materials. We will also open-source both the codebase and the generated dataset. Detailed descriptions of our model architecture, training procedures, and hyperparameters are provided in Section 4.2 of the main paper. Implementation details, including environment setup and dataset details are further elaborated in Section 5.1. The dataset construction pipeline and format are explained in Section 4.1 and Appendix B.

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

# A    CASE STUDY

## A.1    SHOPPING SCENARIO

Figure 3: Successful case 1 by Qwen2.5-VL-7B + CoMEM in shopping scenario. The model accurately understands the goal, retrieves relevant memory, and selects the correct product.

Figure 4: Failure case of the base model Qwen2.5-VL-7B without memory. The model selects an incorrect product that exceeds the price limit, highlighting the importance of contextual memory.

In the task "Find a beginner's acrylic paint set on Amazon, with at least 24 colors, suitable for canvas painting, and priced under $40," the base model exhibits limited reasoning and planning

capabilities as shown in figure 4. It prematurely selects the first product shown, failing to rigorously evaluate whether the item meets all specified constraints such as price, color count, and suitability for canvas painting. This behavior suggests a lack of contextual understanding and strategic planning. In contrast, the CoMEM-enhanced model demonstrates significantly improved performance by leveraging compressed memory representations from multiple prior shopping trajectories, as shown in figure 3. These compact embeddings provide rich context, guiding the model to follow a more deliberate and accurate plan—searching, analyzing multiple candidate products, and validating them against all task constraints before making a final selection. This example highlights how CoMEM enables deeper reasoning and constraint-aware decision-making, leading to more reliable outcomes in complex multi-step tasks.

## A.2 INFORMATION SCENARIO

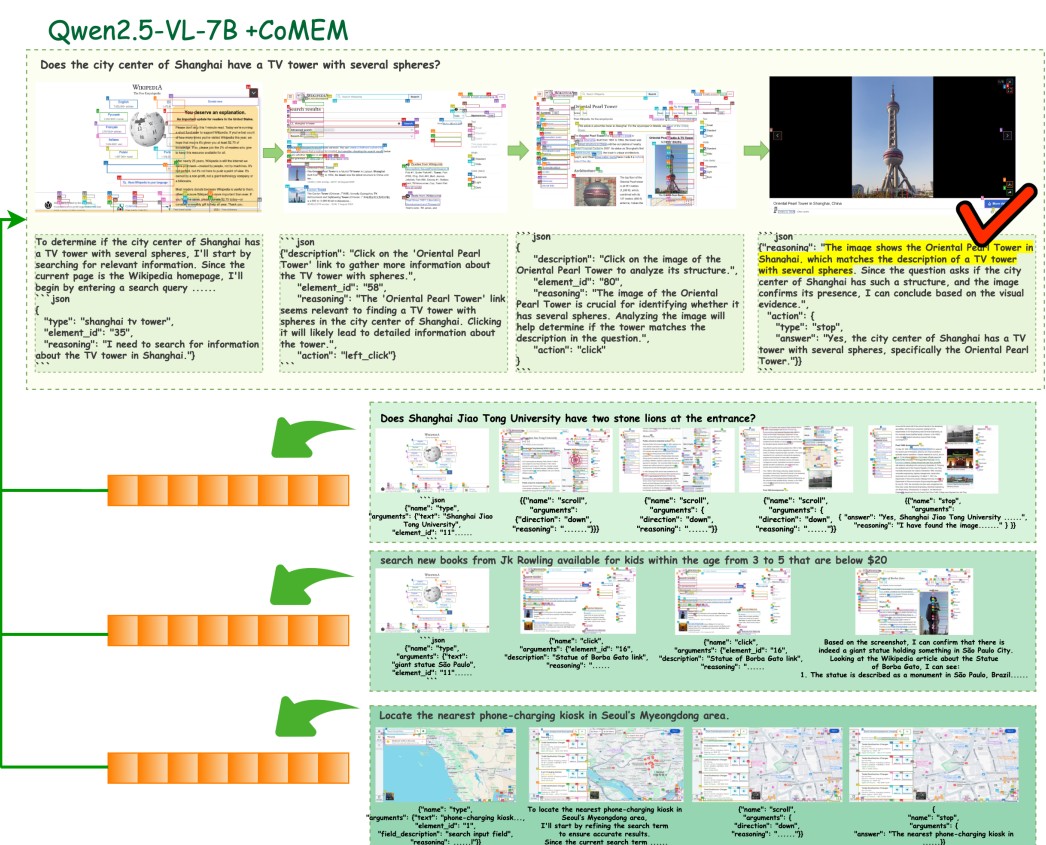

Figure 5: Successful case 2 by Qwen2.5-VL-7B + CoMEM in information scenario. The model accurately understands the goal, retrieves relevant memory, and finds the answer.

In the task "Does the city center of Shanghai have a TV tower with several spheres?", the base model demonstrates inefficient behavior by issuing an overly general query ("Shanghai") and repeatedly scrolling through irrelevant sections of the Wikipedia article. As shown in figure 6, this lack of targeted exploration causes the model to miss the visual and textual cues needed to answer the question, ultimately resulting in failure. In contrast, in figure 5 the CoMEM-enhanced model begins with a more precise search query ("Shanghai TV tower") informed by prior experience. It quickly identifies the Oriental Pearl Tower—whose structure features multiple spheres—and successfully verifies its location using both textual and visual evidence. This comparison highlights how CoMEM enables more focused planning and efficient information retrieval, especially in tasks requiring multimodal reasoning.

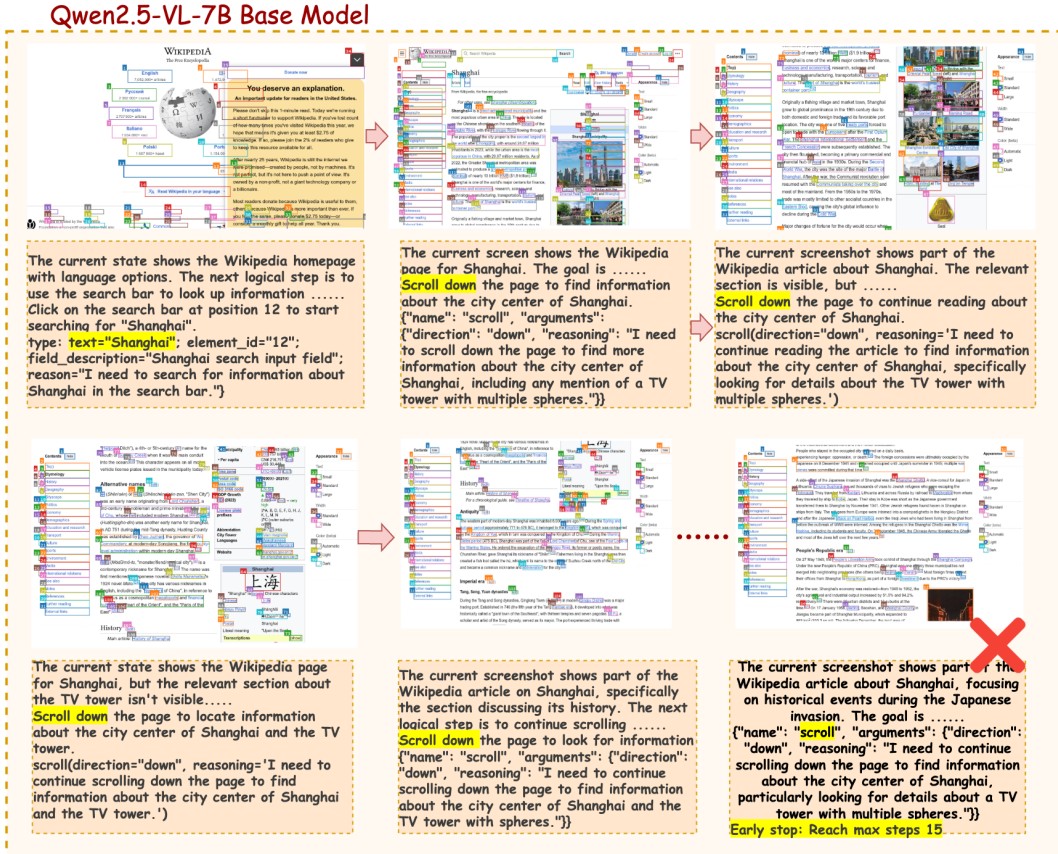

Figure 6: Failure case of the base model Qwen2.5-VL-7B without memory. The model starts with a too general query and lacks enough exploration.

## B  TASK GENERATION PIPELINE

From $\mathcal{Q}_0$, the seed dataset, task descriptions are sampled from each category, of which there are 13 ('education', 'tech', 'entertainment', 'travel', 'health', 'news', 'services','shopping','social', 'food', 'academic', 'government', 'finance').

Each task is used as a search query in DuckDuckGo using SerpAPI, after which the top 20 search results are collected and added to the dictionary of URLs, under the category that the search query was sampled from, resulting in 6,676 unique page links.

As the first part of the task generation process, each link is processed to truncate them down to the parent domain and first path level. The link is opened with Playwright and a screenshot is taken. The link and screenshot are passed to the VLM with the following prompt to extract the page content and check if the page is inaccessible.

---

**Page Content Extraction Prompt**

You are an expert web page analyzer. Analyze the given website and screenshot and extract all useful information in detail. If the page is blocked (e.g. 403, 404, has captcha, etc.), directly return 'blocked'. Otherwise, return a detailed description of the page content.

---

The extracted page content then forms part of the prompt to the VLM to extract detailed information on the page tailored specifically for the task generation LLM's use. At this stage, if a page is blocked, we skip it.

**Page Information Extraction Prompt**

You are an expert web page analyzer. Analyze the given website and screenshot and extract all useful information for task generation.

URL: {url}
Category: {category}
Page Content: {page_content}

Based on the URL, category, and page content above, provide a comprehensive summary of:
1. Website type and purpose - its primary intended use cases, problems that can be solved on this site
2. Navigation structure and key sections
3. Interactive elements (buttons, forms, links, etc.)
4. Content types present (articles, product listings, archives, etc.)
5. Anything unusual or unique to this site
6. IGNORE any cookie notices, privacy popups, log in buttons, site errors, ads, or other elements not specific to the site's function.

Focus on information that would be useful for generating problems for web automation to solve. Think about what human users might want to achieve on this website.
Provide a clear, structured summary that captures these essential aspects of the website for task generation purposes.

The resulting detailed page description forms {info_summary}, part of the task generation prompt. Each prompt also contains {examples_section}, a random selection of 5 example tasks from $\mathcal{Q}_0$, the seed dataset. Examples are pulled from the same task category that the current page belongs to.

**Initial Task Generation Prompt**

You are an expert generator of task problems for web automation. Based on the website analysis and screenshot, generate 10 diverse task problems that a web agent could solve on this website.

URL: {url}
Category: {category}

IMPORTANT: Generate direct, actionable problems with a solution. Task problems should be specific and achievable.

Generate exactly 10 diverse, direct instruction task problems that:
1. Have a clear, specific objective; a problem to solve
2. Are achievable within the website's scope
3. Test diverse skills like navigation, information gathering, information synthesis in order to solve problems
4. Require multiple steps to solve, not a single action
5. Have measurable, verifiable success criteria; avoid vague, unverifiable tasks like 'Read a paragraph' or 'Explore a page'. Instead, focus on a clear, deliberate end goal.
6. IMPORTANT: Do NOT write tasks that contain overly specific instructions like 'Click on X...' or 'Type X...'. These are not problems to be solved. The task problem should not direct the agent on what to do, but rather what to achieve.
7. Are distinct, not related to each other, and do not require knowledge or completion of previous task problems.

For each task problem, provide:
- Task problem description (direct instruction with specific requirements)
- Expected outcome (what should be accomplished, what condition needs to be checked to indicate success)

- Difficulty level (easy/medium/hard)

{examples_section}

Format your response as exactly 10 tasks, one per line, with this structure:
1. [Task Description] | [Expected Outcome] | [Difficulty]
2. [Task Description] | [Expected Outcome] | [Difficulty]
3. [Task Description] | [Expected Outcome] | [Difficulty]

Example format:
1. Find a news article about climate change published in the last week | Should locate and display a recent climate change article | Easy
2. Search for iPhone 12 Pro with price below $800 | Should have navigated to a page for iPhone 12 Pro models under $800 | Medium
3. Book a hotel in New York for 2 people for next weekend | Should find and select a specific hotel for 2 people | Medium
4. Find the top 3 customer reviews for Nike running shoes | Should locate and display a page showing the top 3 reviews | Easy
5. Compare prices of Samsung Galaxy phones between $500-700 | Should find a page comparing multiple Samsung phones in that price range | Medium
6. Find the paper with the most citations under the Computer Science category | Should identify and display the page of an AI paper | Hard

Respond only with the 10 tasks in the specified format, no additional text.

Website Analysis: {info_summary}

The tasks generated by the LLM in this first pass contain some poor-quality tasks that are overly specific and not similar to tasks undertaken by human users.

To refine these tasks, each generated task is passed to the LLM again with a prompt to rewrite the task if necessary. The prompt is based on the one used in ZeroGUI's (Yang et al., 2025) new task generation pipeline.

**Task Refinement Prompt**

You will be given a task instruction. Please refine it with the following requirements:

1. Imitate the speech style of a human user. Make it more natural and diverse.
2. Remove specific hints on how to achieve the task. (e.g. press which button, click which link, etc.)
3. The task should remain unambiguous and clear, and the task's goal must remain the same.

For example, you should refine:
"Use the "Price range" filter on the Ryanair website to limit the flight options to $50-100."
into:
"Find flights from London to Paris with a price range of $50-100."
or
"I only have a limited budget for flights. Could you help find flights from London to Paris with a price range of $50-100."

You should refine:
"Import email messages from another email program by clicking the "Import" button under the "Import from Another Program" section."
into:
"Please import email messages from another email program."
or

"I have some emails in my other email program. Could you help me import them into Thunderbird?"

You should refine:
"Click on the "About Us" link to learn more about the company's history and mission."
into:
"Please provide information about the company's history and mission."
or
"Find an "About Us' or similar page on the website that describes the company's history and mission."

The original instruction is:
{instruction}
Refine it and return the refined instruction text in this exact format:

ORIGINAL: [the original instruction]
REFINED: [the refined instruction]

## THE USE OF LARGE LANGUAGE MODELS

We employed large language models in two limited capacities: first, to polish drafts for grammar, clarity, and flow without changing the scientific content; second, to provide lightweight debugging assistance. All algorithmic designs, experiments, analyses, and claims originate from the authors.

