# OpenReview forum: "Auto-scaling Continuous Memory for GUI Agent"
_ICLR.cc/2026/Conference — Submitted to ICLR 2026_

### Official Review · Reviewer_31Nz · 2025-10-18

**Soundness:** 2
**Presentation:** 2
**Contribution:** 2
**Rating:** 4
**Confidence:** 4

**Summary:**

This paper proposes CoMEM, a framework that enhances training and inference efficiency for GUI agents by integrating memory compression within a self-validating data-generation flywheel. For efficient trajectory representation, each trajectory is encoded into a fixed number (e.g., 8) of learned vectors, which are used as in-context examples and directly injected into the VLM’s input layer. On web-GUI benchmarks, a 7B open-source VLM augmented with this memory matches or surpasses closed-source baselines (e.g., GPT-4o).

**Strengths:**

- The focus on agent scaling with memory is timely for GUI agents. Compressing trajectories while preserving visual cues effectively targets generalization to unseen interfaces and long-horizon tasks.
- The paper presents a simple yet effective scaling approach for GUI agents that uses fixed-length continuous embeddings which can be retrieved and prepended without inflating the context length.
- The system is both effective and efficient, requiring only lightweight fine-tuning of the memory encoder and enabling low-cost memory growth.
- The literature coverage on GUI agents and memory for LLMs is solid, and the proposed pipeline is clearly positioned relative to recent work.

**Weaknesses:**

- The novelty mainly rests on analysis and design choices for GUI agents rather than on a fundamentally new learning principle, as the idea of automating a data collection flywheel through environment interaction is not new for long-horizon sequential decision-making agents [1, 2, 3, 4].
- The paper’s use of “auto-scaling” is potentially misleading. Auto-scaling conventionally denotes elastic compute scaling, whereas the proposed data flywheel increases data, not capacity. The authors should revise or clearly define the term, for example as “automated data expansion” or “automated data augmentation,” to avoid overstating the contribution.
- The approach is validated mainly with a 7B VLM. It assumes open-source VLMs have sufficient planning and verification ability to maintain diversity and quality at scale, but there is no convincing analysis for smaller models. A more practical setting would include transparent results or insights for small models such as 3B and 1B.
- Some numbers are inconsistent between the abstract and the conclusion. The paper lacks a transparent presentation of the collected memory demonstrations, detailed cost accounting, retrieval configuration, and memory bank curation criteria, which hinders reproducibility.
- Analysis and discussion of bottlenecks for scaling the CoMEM framework are limited. The paper asserts that trajectories can be compressed to as few as eight continuous vectors but does not justify when eight are sufficient as trajectory complexity grows. A simple test across complexity regimes would strengthen the claim.
- The authors claim that high-quality training is important, but they do not analyze the robustness or data quality of the memory.
- While the paper presents promising results, the evidence supporting generalization to unseen environments could be strengthened through deeper analysis or focused case studies showing where CoMEM succeeds while the baseline fails, thereby clarifying the factors driving the improvement.

Minor
- Figure 2 is informative but visually dense. Reducing icon clutter and explicitly highlighting the key components of the flywheel would improve readability.

[1] Shinn, Noah, et al. "Reflexion: Language agents with verbal reinforcement learning." Advances in Neural Information Processing Systems 36 (2023): 8634-8652.

[2] Wang, Guanzhi, et al. "Voyager: An open-ended embodied agent with large language models." arXiv preprint arXiv:2305.16291 (2023).

[3] Dai, Zhuyun, et al. “Promptagator: Few-shot Dense Retrieval From 8 Examples.” Proceedings of the International Conference on Learning Representations (ICLR), (2023).

[4] Wang, Zun, et al. “Bootstrapping Language-Guided Navigation Learning with Self-Refining Data Flywheel.” Proceedings of the International Conference on Learning Representations (ICLR), 2025,

**Questions:**

- Could prior knowledge be used to pre-adjust the retrieval range or weighting during evaluation so that the memory focus becomes sharper for a given task family? In particular, how could CoMEM be improved or extended to support such prior-conditioned retrieval?
- For training the memory encoder, what sampling criteria are used to balance the quality and diversity of trajectories? Is there a principled metric for determining the required number of training samples? If fewer but more diverse trajectories are used, could the method achieve better data efficiency? This question arises because performance degrades around 2,000 samples, suggesting sensitivity to sample selection and an increased burden on hyperparameter tuning.

---

> ### Author Response · Authors · 2025-11-20
> **Response to Reviewer 31Nz: part 1**
>
> We sincerely thank the reviewer for the thoughtful and detailed feedback. We appreciate the recognition of the promise in our results and acknowledge the valuable suggestions to help us identify areas to improve clarity and strengthen our claims. Below, we address each of the weaknesses and questions in turn.
>
> ---
> ### **Response to Weakness 1: Distinct Contributions of CoMEM**
>
> We want to clarify that CoMEM introduces a scalable, unified memory framework built on three major innovations, instead of only an automated data collection flywheel:
>
> 1. **A trajectory-level continuous memory architecture** tailored for GUI agents:
>    Our architecture encodes entire multimodal trajectories into compact continuous units (~8 tokens)
>    via *LoRA-trained Q-Former*, enabling efficient injection into agents.
>    This differs markedly from prior step-level scripts or text/image trace banks.
>
> 2. **A closed-loop, auto-scaling memory flywheel**:
>    Unlike prior systems that depend on static, human-annotated datasets, our flywheel automates
>    `Discover → Generate → Rollout → Judge → Store`, enabling large-scale trajectory collection
>    (15,145 steps from 6,676 websites for approx. $553) — achieving unmatched diversity and scale.
>
> 3. **Empirical scaling laws** for memory size and retrieval depth:
>    We are the first to show empirical scaling laws in GUI agents.
>    - Memory bank size increases yield log-linear accuracy gains.
>    - Retrieval depth shows monotonic improvements.
>
> Together, these contributions establish CoMEM as *more than a compression method or data collection method* — it is a scalable, general-purpose memory solution for long-horizon, cross-domain GUI agent tasks.
>
> ---
>
> #### **Comparative Table: CoMEM vs. Prior Memory Approaches**
>
> | Category           | Method               | Modality                 | Scalable | GUI Applicable | Efficient Training                        |
> |--------------------|----------------------|---------------------------|----------|------------------|--------------------------------------------|
> | LM-Memory      | MA-LMM [1]               | Image(+Video)+Text        | Yes      | No               | Unknown data, 200M parameters              |
> | LM-Memory     | VoCo-LLaMA [2]          | Image/Video+Text          | Yes      | No               | 665K data, 7B parameters                    |
> | Workflow Memory| Agent Workflow Memory [3] | Text Task-level guidance       | No       | Yes              | —                                          |
> | Workflow Memory| Reasoning Bank [4]      | Text Task-level guidance       | No       | Yes              | —                                          |
> | Procedural Memory| Memp  [5]       | Text Skill/step reuse          | No       | Yes              | —                                          |
> | This work     | **CoMEM**            | Image+Text (screenshots + actions) | **Yes** | **Yes**         |  1.5K data, 200M parameters |
>
> ---
>
> ### **Response to Weakness 2: Potentially Misleading Term of “Auto-Scaling”**
>
> Thank you for pointing out this important issue regarding terminology. We agree that the term *“auto-scaling”* has a well-established meaning in conventional literature and could be misinterpreted in this context. In our paper, “auto-scaling” was intended to describe two complementary ideas:
>
> - **Automated Data Expansion**:
>   Our proposed data flywheel framework autonomously scales the quantity and diversity of training data by:  1) Discovering new environments;  2) Synthesizing tasks; 3) Collecting trajectories via the agent; 4) Validating them using VLM feedback
>
> This enables the system to grow its memory bank automatically without requiring human annotation — a key enabler for scaling to open-ended environments.
>
> - **Scalability of CoMEM**:
>   The term also referred to the scaling behavior of continuous memory, as shown in Figure 1. Specifically, we demonstrate that increasing the number of retrieved memory embeddings leads to consistent performance improvements — which would not be possible without compact memory representations and automated data collection.
>
> To avoid confusion and align with community expectations, we will *revise the terminology* in the final version of the paper.
>
> ---
>
> [1] Ye, Xubing, et al. “VoCo-LLaMA: Towards Vision Compression with Large Language Models.” arXiv preprint arXiv:2406.12275 (2024).
>
> [2] He, Bo, et al. “MA-LMM: Memory-Augmented Large Multimodal Model for Long-Term Video Understanding.” arXiv preprint arXiv:2404.05726 (2024).
>
> [3] Wang, Zhiruo, et al. “Agent Workflow Memory.” arXiv preprint arXiv:2409.07429 (2024).
>
> [4] Ouyang, Siru, et al. “ReasoningBank: Scaling Agent Self-Evolving with Reasoning Memory.” arXiv preprint arXiv:2509.25140 (2025).
>
> [5] Fang, Runnan, et al. “Memp: Exploring Agent Procedural Memory.” arXiv preprint arXiv:2508.06433 (2025).

---

> ### Author Response · Authors · 2025-11-20
> **Response to Reviewer 31Nz: part 2**
>
> ### **Response to Weakness 3: Effectiveness of Method on Smaller Model**
>
> **GUI Reasoning Remains Challenging for Small Models**
>
> First, it’s important to note that GUI-based tasks involve a combination of long-horizon planning, error correction, layout understanding, and step-by-step execution — all of which stretch the capabilities of current VLMs. As shown in Table 2 of our paper, even the 7B Qwen2.5-VL struggles with many of these tasks. This challenge is more serious for smaller models (e.g., 3B or 1B), which often fail to complete multi-step tasks without external guidance.
>
> **CoMEM Performance on Qwen2.5-VL 3B**
>
> To address this concern, we conducted additional experiments using *Qwen2.5-VL-3B*. We trained a memory encoder for the 3B model and evaluated performance **with and without CoMEM**. As shown in the table below, CoMEM significantly boosts performance even for a 3B model — in some cases by more than 15 percentage points — despite the model’s limited reasoning capacity.
>
> Therefore we can conclude that:
>
> - CoMEM is generalizable to smaller models.
> - The memory encoder effectively compensates for limited internal capacity by injecting distilled, actionable knowledge from prior experience.
> - Even when the base model struggles, CoMEM provides task-relevant priors that improve decision making.
>
>
> **Performance Table (Qwen2.5-VL-3B)**
>
> | Model         | MMInA-Wikipedia | Mind2Web-Travel | Mind2Web-Info  | Mind2Web-Service | Webvoyager |
> |---------------|-----------|--------|--------|---------|------------|
> | Qwen2.5-VL-3B | 0.00%     | 0.00%  | 1.67%  | 1.96%   | 9.80%      |
> | + text-memory   | 1.60%     | 4.16%  | 10.53% | 4.92%   | 14.29%     |
> | + CoMEM   | 4.22%     | 8.33%  | 12.68% | 8.20%   | 27.27%     |
>
> ---
>
> ### **Response to Weakness 4: Details for Memory Construction and Demonstration**
>
> **Retrieval configuration.**
>
> At inference time, given the current observation $o_t$ (a screenshot), we perform embedding-based multimodal retrieval over previously stored trajectories. Concretely, for each stored trajectory, *CLIP* (a contrastive language–image pretraining model that maps images and text into a shared embedding space so semantically matched pairs lie close under cosine similarity) encodes every step-level screenshot together with its associated action/query; the resulting embeddings are pooled into a single multimodal key per trajectory and indexed with *FAISS* (a high-performance library for nearest-neighbor search over large vector collections that supports efficient exact or approximate retrieval). The agent queries this index with the embedding of the current $o_t$ and retrieves the top-$k$ nearest neighbor trajectories. In our scaling studies, we vary the number of retrieved items $k$ and observe consistent gains for continuous memories.
>
> **Memory-bank curation criteria.**
>
> A dedicated *judge VLM (SEAgent-1.0-7B [1])* evaluates each completed trajectory by reading the task query plus the full step-wise trajectory and deciding success under the standard LLM-as-Judge criterion. Only judge-verified successful trajectories, together with their task queries and environments, are admitted to the memory bank and update the trajectory/task/environment pools. To keep the bank reliable and useful for retrieval, we enforce three practical rules during ingestion:
> - (i) **Stability Screening** at the environment level (filter sites that fail basic accessibility checks or exhibit non-deterministic behavior);
> - (ii) **Redundancy Control** via de-duplication at the parent-domain level and removal of near-duplicate trajectories (measured by screenshot/action embedding similarity);
> - (iii) **Diversity Balancing** across environment categories to mitigate over-concentration on “easy” layouts.
>
> **Detailed cost accounting.**
>
> For website expansion, we used the free tier of SerpAPI. For task generation and trajectory rollout, we accessed VLMs via the OpenRouter API. The pricing is as follows: Qwen2.5VL-7B — \\$0.20/M input/output tokens; Qwen2.5VL-32B — \\$0.05/M input, $0.22/M output; UI-TARS-7B — \\$0.10/M input, \\$0.20/M output. The total cost of data collection was \\$553, covering all OpenRouter API usage.
>
> **Collected memory demonstrations.**
>
> Our continuous memory bank spans 16 categories—education, technology, entertainment, travel, health, etc. In `trajectory_evaluation` folder of supplementary file, we provide representative GUI interaction trajectories from the bank. Each example presents step-by-step screenshots aligned with the agent’s actions.

---

> ### Author Response · Authors · 2025-11-20
> **Response to Reviewer 31Nz: part 3**
>
> ### **Response to Weakness 5: Justifications for using 8 tokens embedding**
>
> **Goal: Compression for Generalization, Not Lossless Copy**
>
> Our objective is to distill *decision-relevant abstractions* from each trajectory, not to preserve every low-level detail. The memory encoder (VLM + Q-former) is trained to summarize raw trajectories into compact, high-utility embeddings that support robust planning across task variations.
>
> **Empirical Ablation on Token Count**
>
> We conducted ablations on the *MMInA shopping domain*, comparing memory encoders trained with 4, 8, and 16 token embeddings per trajectory. The number of retrieved memories was varied from 0 to 100.
>
> - *4 tokens*: Too compressed; task-relevant information is lost. Performance saturates quickly.
> - *16 tokens*: Performs well with few memories, but degrades as more are added due to input length saturation. This is likely because the memory encoder was trained with only 3 samples, so it doesn’t generalize well to 30+ samples. We believe this is a result of training–inference mismatch caused by compute constraints. With more training samples, the 16-token variant could likely also exhibit scaling behavior.
> - *8 tokens*: Achieves *strong performance with consistent scaling*, supporting more retrieved memories without degradation.
>
> | Retrieved Memories         | 0     | 3     | 10    | 20    | 30    | 50    | 70    | 90    | 100   |
> |-----------------------------|-------|-------|-------|-------|-------|-------|-------|-------|--------|
> | Text-Memory                 | 15.5% | 22.0% | 30.5% | 31.0% | 29.0% | 25.0% | 21.0% | 22.0% | 21.5%  |
> | CoMEM (4 tokens)   | 15.5% | 30.5% | 32.0% | 35.0% | 37.0% | 38.5% | 37.5% | 39.0% | 40.0%  |
> | CoMEM (8 tokens)   | 15.5% | 41.5% | 45.0% | 43.0% | 42.0% | 43.5% | 44.5% | 45.5% | 46.0%  |
> | CoMEM (16 tokens)  | 15.5% | 42.5% | 46.5% | 47.0% | 44.0% | 41.5% | 38.5% | 39.0% | 38.0%  |
>
> Therefore, we conclude that using more tokens allows the model to store more information. In our experiments, 8 tokens are sufficient to distill high-level, decision-relevant information from trajectories of up to 15 steps. However, for longer or more complex trajectories, using more tokens may be beneficial to capture the additional procedural details and information.
>
> ---
>
> ### **Response to Weakness 6: Robustness on Memory Quality**
>
> In this paper, we use CLIP+FAISS as the first-stage retriever, then apply several additional filters to ensure retrieval quality. These steps already provide a reasonable level of robustness, as evidenced by the strong performance of CoMEM reported in our main results:
>
> - *Multimodal Retrieval*: We compute similarity between the current task (image + text) and the initial screenshot + query from each candidate trajectory.
> - *Domain-Aware Filtering*: Each task is tagged with a domain label (e.g., shopping, travel, info), and we prioritize memory items from the same domain to reduce semantic drift.
> - *Trajectory-Level Quality Control*: Only successful trajectories are stored in memory, and we prune noisy or redundant steps using rule-based heuristics during memory construction.
>
> Besides, our continuous memory design accepts any retrieved trajectory and compresses it into a fixed-length embedding block. It means that more sophisticated retrieval pipelines (e.g., cross-encoder re-ranking) can be plugged in without modifying the memory structure or the agent architecture. We view this flexibility as one of CoMEM’s key strengths.
>
> To directly address the reviewer’s suggestion, we implemented a VLM-based quality gating mechanism as a lightweight second-stage filter. Specifically, we:
>
> - Retrieve the top-30 candidates using CLIP+FAISS.
> - Use a VLM to judge each candidate trajectory’s relevance and quality for the current task.
> - Keep only the top-10 filtered items to encode into memory embeddings.
>
> As shown in the table below, this simple gating mechanism leads to consistent performance improvements across domains. This confirms that retrieval quality is indeed important, and that CoMEM benefits from stronger reranking — a promising direction for future work.
>
> Notably, we also performed a *stress test* by introducing noisy memory trajectories into the retrieval pool. As shown in the last row, while performance drops slightly compared to clean memory, *CoMEM still maintains significant gains over the base agent*, highlighting the robustness of our continuous memory design.
>
> **CoMEM Stress Test and Quality Gating Results**
>
> | Model           | Wikipedia | Travel | Info  | Service | Webvoyager |
> |--------------------------|-----------|--------|-------|---------|------------|
> | Qwen2.5-VL                | 36.7%     | 9.5%   | 9.6%  | 17.3%   | 40.0%      |
> | + CoMEM                  | 47.4%     | 18.8%  | 26.5% | 17.7%   | 54.5%      |
> | + VLM gating + CoMEM     | 48.3%     | 19.6%  | 28.6% | 19.6%   | 56.5%      |
> | + Noisy Memory + CoMEM   | 44.7%     | 17.1%  | 21.7% | 16.7%   | 49.0%      |

---

> ### Author Response · Authors · 2025-11-20
> **Response to Reviewer 31Nz: part 4**
>
> ### **Response to Weakness 7: Generalization to Unseen Environments**
>
> Firstly, **as shown in Table 3 of our paper, we evaluated the generalization ability of our method to unseen desktop/app environments**. These experiments demonstrate that even when memory is collected from a completely different domain (web navigation) and applied to desktop/app environments (e.g., Word, PowerPoint, VS Code), our continuous memory method (CoMEM) still improves performance over the baseline—achieving an *average absolute gain of +1.6%*. This suggests that CoMEM captures *abstract, transferable patterns* of GUI interaction (e.g., multi-step planning, grounding UI elements, feedback loops) that are *not specific to any single domain*.
>
> To explore whether the performance of CoMEM can be better with *domain aligned memory*, we conducted additional experiments using *domain-aligned memory*. Specifically, we used *13,750 trajectories* from the training set of [GUI-360°](https://arxiv.org/abs/2511.04307) [1] as our memory bank, a dataset covering desktop applications like Microsoft Word, Excel, and PowerPoint.
>
> When memory is drawn from this desktop/app domain, we observe *substantially higher gains* on both *GUI-Oddsey* and *OSWorld*. These results are summarized in the table below and demonstrate that our framework generalizes well across domains—*especially when memory is matched appropriately*.
>
> Moreover, our *auto-scaling data flywheel* plays a pivotal role in bridging domain gaps. Unlike prior work that relies on static, human-annotated datasets, our framework supports the *continuous collection of high-quality trajectories* from arbitrary domains—including desktop applications. This flexibility allows our agent to adapt quickly to novel environments and build *general-purpose memory banks* that span diverse UI paradigms.
>
>
> **Unseen Environment Generalization Results**
>
> | Memory Type               | GUI-Oddsey (AMS) High-Level | GUI-Oddsey (AMS) Low-Level | OSWorld (SR) Office | OSWorld (SR) Daily | OSWorld (SR) Professional | OSWorld (SR) Overall | Avg (%) |
> |---------------------------|-----------------------------|-----------------------------|---------------------|--------------------|----------------------------|----------------------|---------|
> | **Baseline**              | 22.4%                       | 45.6%                       | 24.7%               | 25.6%              | 60.2%                      | 26.4%                | 35.7    |
> | Web-domain + Text Memory  | 24.4%                       | 37.4%                       | 23.1%               | 23.1%              | 57.1%                      | 24.7%                | 33.0    |
> | Web-domain + CoMEM        | 27.4%                       | 44.9%                       | 25.1%               | 28.2%              | 60.9%                      | 26.7%                | 37.3    |
> | Desktop-domain + Text Mem | 26.9%                       | 48.8%                       | 27.3%               | 28.2%              | 63.3%                      | 27.8%                | 38.9    |
> | **Desktop-domain + CoMEM**| **31.0%**                   | **52.2%**                   | **30.7%**           | **30.8%**          | **67.4%**                  | **30.0%**            | **42.4** |
>
> [1] Liu, Zihan, et al. “GUI-360°: A Benchmark for Multi-Platform GUI Agents.” arXiv preprint arXiv:2511.04307 (2025).

---

> ### Author Response · Authors · 2025-11-20
> **Response to Reviewer 31Nz: part 5**
>
> ### **Response to Question 1. Could prior knowledge be used to improve CoMEM?**
>
> Thank you for the thoughtful suggestion. Incorporating prior knowledge to guide memory retrieval is a promising direction. To study it, we conduct an ablation to evaluate CoMEM’s robustness to memory quality and its benefits from more selective retrieval.
>
> - *+ VLM Gating*: Retrieved memory is filtered using a vision-language model to retain only top-10 high-quality, relevant trajectories.
> - *+ Noisy Memory*: We inject 50% irrelevant memory (lowest-5 similarity) to test degradation.
>
> According to the results in the table below, *CoMEM is robust to moderate noise* due to its high-quality memory bank construction, and *prior-informed filtering (e.g., VLM gating)* further improves performance.
>
> | Model                         | MMInA-Wikipedia |	Mind2Web-Travel |	Mind2Web-Info |	Mind2Web-Service |	Webvoyager |
> |------------------------------|-----------|--------|--------|---------|------------|
> | Qwen2.5-VL                   | 36.7%     | 9.5%   | 9.6%   | 17.3%   | 40.0%      |
> | + CoMEM                      | 47.4%     | 18.8%  | 26.5%  | 17.7%   | 54.5%      |
> | + VLM gating + CoMEM         | **48.3%** | 19.6%  | 28.57% | 19.6%   | 56.5%      |
> | + Noisy Memory + CoMEM       | 44.7%     | 17.1%  | 21.7%  | 16.7%   | 49.0%      |
>
> ### **Response to Question 2: Data Efficiency for Training**
>
> Thank you for the thoughtful question. Our approach to training the memory encoder emphasizes balanced action coverage and domain diversity, while maintaining data efficiency due to the nature of the task.
>
> ---
>
> #### **Sampling Strategy**
> We trained the memory encoder using 1,000 trajectories from the Mind2Web training set, and 500 trajectories collected via our pipeline. These span 13 diverse domains (e.g., shopping, social media, travel), with step-level supervision (ground-truth actions per step).
>
> To ensure balanced coverage, we controlled for action type distribution, maintaining roughly uniform representation across common action types (e.g., click, type, scroll).
>
> ---
>
> #### **Training Efficiency and Diversity**
> As described in Section 5.3, the goal is to align the compressed memory embeddings with VLM representations, not to learn a new task from scratch. Since we use the same VLM for encoding and inference, the memory encoder:
>
> - Converges quickly
> - Is not data-hungry
> - Needs to avoid overfitting, especially given the small token budget
>
> Thus, we prioritized action and domain diversity over raw sample size. In fact, we observed that with ~2,000 samples, performance slightly degrades — likely due to redundancy and overfitting in fine-tuning. This suggests that fewer but more diverse samples (for example, 1500 training samples as used in our paper) can indeed improve data efficiency.
>
> ---
>
> #### **Broader Data Diversity via the Memory Flywheel**
> While the training set for the encoder is small, **the real diversity comes from the memory bank itself**, powered by our automated data flywheel (discover → generate → rollout → verify). This ensures that the injected memories during inference are rich and varied, which has a larger impact on downstream performance than the encoder data alone.

---

> ### Comment · Reviewer_31Nz · 2025-11-27
>
> I appreciate the authors for addressing the reviewers’ comments thoroughly. After reading the rebuttal, the additional experiments, and the discussions among reviewers, I have updated my score. The authors incorporated the suggestions effectively, and the expanded analysis appear well aligned with the concerns raised. In particular, the analytical contributions seem valuable and, in my view, will provide a useful foundation for the future development of GUI agents, especially with respect to memory systems.
>
> ---
>
> That said, a few limitations and concerns remain unresolved. The paper combines several existing ideas and aims to provide a framework for GUI agents that could, in principle, be extended to other domains. As a result, there is an inherent difficulty in clearly articulating the connections across these broader research directions. In addition, although the authors present ablation studies demonstrating robustness to hyperparameters, the method still faces a fundamental challenge: it is difficult to define generally applicable tuning strategies across diverse datasets and task distributions.
> This issue is particularly relevant for GUI agents, where even the same website or application can change frequently due to rapid UI updates, layout drift, and evolving interaction patterns. Such variability amplifies the difficulty of defining generally applicable tuning strategies and retrieval configurations, and this remains insufficiently addressed within the current scope. Nonetheless, within the boundaries of the presented experiments, the authors have made reasonable efforts, and these limitations alone do not justify rejecting the paper.
>
> Overall, I believe the analytical and empirical contributions outweigh the remaining concerns, and I am leaning toward the accept side. I would also encourage the authors to ensure that the discussed revisions and experimental updates are clearly reflected in the final version, as it is currently difficult to verify where these changes have been incorporated.

---

### Official Review · Reviewer_BtWp · 2025-10-28

**Soundness:** 2
**Presentation:** 2
**Contribution:** 1
**Rating:** 2
**Confidence:** 4

**Summary:**

This paper proposes a continuous memory methodology to address the generalization challenges GUI agents face with unfamiliar interfaces and long-horizon tasks. Existing GUI agents compress past trajectories into text tokens, which dramatically increases context length and loses crucial visual cues (e.g., exact widget size and position).

To address this, the authors propose CoMEM (Continuous Memory). Each GUI trajectory is compressed into a fixed-length sequence of 8 continuous embeddings using the VLM itself as an encoder, which are then directly injected into the backbone's input layer. Experimental results demonstrate that continuous memory shows monotonic performance improvement as memory size and retrieval depth increase, whereas text-based memories degrade beyond roughly ten retrieved items due to ballooning sequence length, increased attention overhead, and accumulated semantic noise.

The authors also propose an auto-scaling data flywheel for memory scaling. This pipeline consists of four stages: (1) discovering new environments via search engines, (2) synthesizing tasks with open-source VLMs, (3) rolling out trajectories with the agent model, and (4) verifying success with the VLM. Using this pipeline, they collected 15,145 GUI trajectories for approximately $553 and fine-tuned only the memory encoder (LoRA on Q-Former, 1.2% of parameters) with 1,500 training samples.

In benchmark experiments, Qwen-2.5-VL-7B equipped with CoMEM achieves performance comparable to or better than state-of-the-art closed-source models like GPT-4o and Claude-4, significantly outperforming them on the Webvoyager dataset. The paper demonstrates that this scaling law can be leveraged to continuously improve GUI agent performance through low-cost, automated data collection and efficient training.

**Strengths:**

- The paper is well-motivated and easy-to-read.
- The auto-scaling data flywheel presents a practical method to collect large-scale data (15,145 trajectories) at low cost ($553) without human annotation.

**Weaknesses:**

- The differences from existing memory-based methodologies are not clearly visible. It seems necessary to clearly articulate the differences from various approaches that enhance GUI Agents using memory.
-  The only difference from existing methodologies in the paper appears to be compressing existing trajectories through a memory encoder. The contribution seems insufficient.
- The baselines only include Text-based Memory, which is a very naive memory utilization method, without including other memory-based approaches. Recent baselines such as AWM or Memp should be included.
- This paper does not include ablation experiments..

**Questions:**

- What exactly are the criteria for retrieving trajectories?
- Qwen2.5-VL-7B shows higher performance than UI-TARS-1.5-7B. what is the reason for this?

---

> ### Author Response · Authors · 2025-11-20
> **Response to Reviewer BtWp: part1**
>
> We sincerely thank the reviewer for the thoughtful and constructive feedback. We appreciate the opportunity to clarify the novelty of our memory design, justify the baselines and methodology, and respond to the specific concerns raised. Below, we address each point in detail.
>
> ---
> ### **Response to Weakness1 & 2: Distinct Contributions of CoMEM**
>
> CoMEM introduces a scalable, unified memory framework built on three major innovations:
>
> 1. **A closed-loop, auto-scaling memory flywheel**:
>    Unlike prior systems that depend on static, human-annotated datasets, our flywheel automates
>    `Discover → Generate → Rollout → Judge → Store`, enabling large-scale trajectory collection
>    (15,145 steps from 6,676 websites for approx. $553) — achieving unmatched diversity and scale.
>
> 2. **Empirical scaling laws** for memory size and retrieval depth:
>    We are the first to show empirical scaling laws in GUI agents.
>    - Memory bank size increases yield log-linear accuracy gains.
>    - Retrieval depth shows monotonic improvements.
>
>    This contributes new empirical insight well beyond simple compression.
>
> 3. **A trajectory-level continuous memory architecture** tailored for GUI agents:
>    Our architecture encodes entire multimodal trajectories into compact continuous units (~8 tokens)
>    via *LoRA-trained Q-Former*, enabling efficient injection into agents.
>    This differs markedly from prior step-level scripts or text/image trace banks.
>
> Together, these contributions establish CoMEM as *more than a compression method* — it is a scalable, general-purpose memory solution for long-horizon, cross-domain GUI agent tasks.
>
>
> ### Comparative Table: CoMEM vs. Prior Memory Approaches
>
> | Category           | Method               | Modality                 | Scalable | GUI Applicable | Efficient Training                        |
> |--------------------|----------------------|---------------------------|----------|------------------|--------------------------------------------|
> | LM-Memory      | MA-LMM [1]               | Image(+Video)+Text        | Yes      | No               | Unknown data, 200M parameters              |
> | LM-Memory     | VoCo-LLaMA [2]          | Image/Video+Text          | Yes      | No               | 665K data, 7B parameters                    |
> | Workflow Memory| Agent Workflow Memory [3] | Text Task-level guidance       | No       | Yes              | —                                          |
> | Workflow Memory| Reasoning Bank [4]      | Text Task-level guidance       | No       | Yes              | —                                          |
> | Procedural Memory| Memp  [5]       | Text Skill/step reuse          | No       | Yes              | —                                          |
> | This work     | **CoMEM**            | Image+Text (screenshots + actions) | **Yes** | **Yes**         |  1.5K data, 200M parameters |
>
> [1] Ye, Xubing, et al. “VoCo-LLaMA: Towards Vision Compression with Large Language Models.” arXiv preprint arXiv:2406.12275 (2024).
>
> [2] He, Bo, et al. “MA-LMM: Memory-Augmented Large Multimodal Model for Long-Term Video Understanding.” arXiv preprint arXiv:2404.05726 (2024).
>
> [3] Wang, Zhiruo, et al. “Agent Workflow Memory.” arXiv preprint arXiv:2409.07429 (2024).
>
> [4] Ouyang, Siru, et al. “ReasoningBank: Scaling Agent Self-Evolving with Reasoning Memory.” arXiv preprint arXiv:2509.25140 (2025).
>
> [5] Fang, Runnan, et al. “Memp: Exploring Agent Procedural Memory.” arXiv preprint arXiv:2508.06433 (2025).

---

> ### Author Response · Authors · 2025-11-20
> **Response to Reviewer BtWp: part2**
>
> ### **Response to Weakness 3: Text-based Memory Baseline**
>
> We implemented two advanced text-memory baselines inspired by recent literature to address your concern:
>
> - *Agent Workflow Memory (AWM)* [1]: Induces reusable workflow templates from successful trajectories and injects them into the prompt.
> - *Reasoning Bank* [2]: Extracts memory from both successful and failed trajectories, allowing the agent to learn from both experience and reflection.
>
> These methods represent stronger and more structured approaches to text-based memory. All implementations use the same retrieval setup as CoMEM and are injected into the prompt using carefully designed templates. As shown in the result table, these structured memory approaches provide modest improvements in some domains but remain inconsistent and significantly underperform CoMEM. This clearly shows two key points:
>
> - Text-based memory is indeed more brittle and variable due to reliance on prompt formatting and token limits.
> - CoMEM offers robust and scalable gains by leveraging compact, multimodal embeddings that generalize better across domains.
>
> ### Table: Performance Comparison Across Domains
>
> | Method                     | MMInA-Wikipedia | Mind2Web-Travel | Mind2Web-Info  | Mind2Web-Service | Webvoyager |
> |---------------------------|-----------|--------|-------|---------|------------|
> | Qwen2.5-VL                | 36.7%     | 9.5%   | 9.6%  | 17.3%   | 40.0%      |
> | + Text-based Memory       | 34.2%     | 17.8%  | 12.7% | 16.6%   | 44.0%      |
> | + Agent-Workflow-Memory     | 37.7%    | 12.5%  | 14.4% | 17.2%   | 44.0%     |
> | + Reasoning Bank            | 37.3%    | 12.0%  | 15.5% | 16.4%   | 41.2%     |
> | + CoMEM                   | **47.4%**     | **18.8%**  | **26.5%** | **17.7%**   | **54.5%**      |
>
> [1] Wang, Zhiruo, et al. “Agent Workflow Memory.” arXiv preprint arXiv:2409.07429 (2024).
>
> [2] Ouyang, Siru, et al. “ReasoningBank: Scaling Agent Self-Evolving with Reasoning Memory.” arXiv preprint arXiv:2509.25140 (2025).

---

> ### Author Response · Authors · 2025-11-20
> **Response to Reviewer BtWp: part 3**
>
> ### **Response to Weakness 4: Need More Ablation Experiments Results**
>
> Because our method is natural and integrated, our original submission emphasized main results, large-scale evaluations, and some further analysis. Below, we provide three focused ablation analyses to address this concern.
>
> ---
>
> ### Ablation 1: Importance of Multimodal Signals in Memory Embeddings
>
> We ablate the role of visual and textual inputs in the memory encoder by disabling one modality at a time. Results below show that both image and text are critical for performance:
>
> *Text offers semantic cues about user intent and actions, while images provide spatial and visual context. Removing either modality significantly degrades performance, validating our choice of fully multimodal memory embeddings.*
>
> |                          | MMInA-Wikipedia | Mind2Web-Travel | Mind2Web-Info  | Mind2Web-Service | Webvoyager |
> |--------------------------|-----------|--------|-------|---------|-------------|
> | Qwen2.5-VL               | 36.7%     | 9.5%   | 9.6%  | 17.3%   | 40.0%       |
> | + CoMEM                  | 47.4%     | 18.8%  | 26.5% | 17.7%   | 54.5%       |
> | + CoMEM without text     | 38.8%     | 12.5%  | 10.2% | 17.0%   | 45.8%       |
> | + CoMEM without image    | 37.5%     | 11.1%  | 12.0% | 17.0%   | 45.3%       |
>
> ---
>
> ### Ablation 2: Robustness to Memory Quality and Retrieval Noise
>
> We examine CoMEM’s sensitivity to retrieval quality:
>
> - *+ VLM Gating*: Retrieved memory is filtered using a vision-language model to retain only top-10 high-quality, relevant trajectories.
> - *+ Noisy Memory*: We inject 50% irrelevant memory (lowest-5 similarity) to test degradation.
>
> *We find that CoMEM is robust to some degree of noise, because during the construction of the memory bank, we have controlled the quality of trajectories. VLM-based gating further improves performance, showing that memory quality matters. As our design is integrated, we can easily combine our method with other more advanced retrieval methods, to make the performance better.*
>
> |                                | MMInA-Wikipedia | Mind2Web-Travel | Mind2Web-Info  | Mind2Web-Service | Webvoyager |
> |--------------------------------|-----------|--------|--------|---------|-------------|
> | Qwen2.5-VL                     | 36.7%     | 9.5%   | 9.6%   | 17.3%   | 40.0%       |
> | + CoMEM                        | 47.4%     | 18.8%  | 26.5%  | 17.7%   | 54.5%       |
> | + VLM gating + CoMEM           | 48.3%     | 19.6%  | 28.57% | 19.6%   | 56.5%       |
> | + Noisy Memory + CoMEM         | 44.7%     | 17.1%  | 21.7%  | 16.7%   | 49.0%       |
>
> ---
>
> ### Ablation 3: Memory Size and Retrieval Count (Scaling Analysis)
>
> As shown in **Figure 1** of the main paper, we ablate the number of trajectories retrieved (from 3 to 100) and the size of the memory bank.
>
> - Results show a clear scaling trend — more memory leads to better performance, up to saturation.
> - This supports our claim that external memory enables continual performance scaling, especially when paired with compression that avoids prompt bloat.

---

> ### Author Response · Authors · 2025-11-20
> **Response to Reviewer BtWp: part4**
>
> ### **Response to Question 1: Detailed criteria for retrieving trajectories**
>
> Below is a precise description of how trajectories are retrieved and the criteria we enforce to ensure retrieval quality and robustness.
>
>
> #### **First-stage retrieval (CLIP + FAISS):** Given the current observation and its task/query text, we build a multimodal query embedding with CLIP, which projects images and short text into a shared space. Every stored trajectory has step-level screenshot/text embeddings that are pooled into a single multimodal key per trajectory. We L2-normalize keys and index them with FAISS; at inference we fetch the top-k nearest neighbor trajectories by cosine/inner-product similarity to the current query embedding. This is the same mechanism described in the paper’s “Memory Retrieval.”
>
> #### **Multimodal similarity scoring:** Similarity is computed jointly from image + text: the query uses the current screenshot and task instruction, and each candidate uses its pooled screenshot/action-query embeddings. Pooling across steps ensures that matches reflect both visual layout and task semantics rather than a single frame, which the paper formalizes as mapping “screenshots and associated actions/queries” to embeddings and pooling them into one trajectory key.
>
> #### **Domain-aware filtering:** Each task/trajectory carries a domain tag (e.g., shopping, travel, information). After the first-stage retrieval, we prioritize same-domain items and down-weight cross-domain candidates. This follows the paper’s stated use of diversity filters during memory construction and its recommendation of domain-balanced crawling/retrieval to avoid semantic drift and self-reinforcement on “easy” sites/layouts.
>
> #### **Trajectory-level quality control:** Only judge-verified successful trajectories are eligible for retrieval: the data flywheel’s Phase-4 “Quality Checking” feeds the task and full trajectory to a dedicated judge VLM (SEAgent-1.0-7B); only positives (successes) are inserted into the memory and used to update the task/environment/trajectory pools. This restricts retrieval to high-quality, successful demonstrations. In addition, the pipeline applies simple diversity and quality filters to prevent redundancy and improve robustness, which implicitly prunes noisy/near-duplicate material before it can be retrieved.
>
> ---
>
> ### **Response to Qestion 2: Reasons for Why Qwen Outperforms UI-TARS**
>
> According to our evaluation and assessment, the performance gap between Qwen2.5-VL-7B and UI-TARS-1.5-7B can be attributed to differences in *model backbone*, *training data emphasis*, and *task alignment*.
>
> #### **Model Design and Training Objectives**
>
> *UI-TARS-1.5-7B* is a continually trained variant of *Qwen-2.5-VL (7B)*, optimized for *GUI grounding and perception*. It is trained on ~50B tokens with a strong focus on:
>
> - Screenshot understanding (element descriptions, dense captioning, state transitions)
> - Coordinate-grounded action prediction
> - Pixel-level alignment for desktop GUI tasks
>
> These design choices lead to state-of-the-art grounding accuracy (e.g., in ScreenSpot-Pro) and strong performance on platforms like OSWorld and AndroidWorld. However, our benchmark emphasizes *web-based*, *long-horizon*, and *goal-directed planning tasks*, which differ significantly from the *desktop GUI* focus of UI-TARS.
>
> #### **Qwen2.5-VL-7B’s Advantages in Our Setting**
>
> *Qwen2.5-VL-7B* benefits from a *stronger general-purpose backbone*, with broader multimodal pretraining and alignment. It targets multimodal reasoning, instruction-following, and planning, making it better suited for:
>
> - Complex web navigation
> - Dynamic page layouts
> - Long-term decision-making
>
>
> #### **Case Studies: Where UI-TARS Falls Short**
>
> To further illustrate the differences, we include three representative failure cases where CoMEM with Qwen2.5-VL-7B succeeds but UI-TARS fails. You can find the trajectory records in the `case_study_comparision` folder in the supplementary file:
>
> - *MMInA_Wikipedia_74 – Replanning Failure*:
>   UI-TARS initially opens a correct page but gets stuck after encountering an unhelpful image. It fails to replan or backtrack, despite prompt instructions allowing retries.
>
> - *MMInA_Shopping_63 – Prompt Ignorance*:
>   UI-TARS disregards the task instruction, randomly clicks a product, and fails to execute the goal. This reflects weak alignment with instruction-following.
>
> - *Webvoyager_Coursera_4 – Random Clicking*:
>   UI-TARS repeatedly clicks the search button without entering a query. We observe this behavior frequently, likely due to training bias toward low-effort click actions.
>
> These examples highlight *systematic limitations* in UI-TARS when applied to web-based, planning-intensive tasks, confirming that the performance gap is rooted in *architectural and training differences*.

---

> > ### Comment · Reviewer_BtWp · 2025-11-27
> >
> > I appreciate the authors' detailed responses. Most of my concerns have been addressed, though I still have some remaining questions regarding the contribution. The authors' claim that "We are the first to show empirical scaling laws in GUI agents" appears to be valid; however, this contribution seems rather narrow and highly specific to the GUI agents domain. For instance, works such as [1][2] have already extensively addressed scaling laws for retrieval in the broader context. Nevertheless, since most of my initial concerns have been resolved, I will adjust my score accordingly.
> >
> > [1] Fang, Yan, et al. "Scaling laws for dense retrieval." Proceedings of the 47th International ACM SIGIR Conference on Research and Development in Information Retrieval. 2024.
> >
> > [2] Shao, Rulin, et al. "Scaling retrieval-based language models with a trillion-token datastore." Advances in Neural Information Processing Systems 37 (2024): 91260-91299.

---

> > > ### Author Response · Authors · 2025-11-27
> > >
> > > Thank you for your thoughtful response and your willingness to consider raising the score — we truly appreciate it! We’re grateful for your constructive feedback, and we would like to take this opportunity to further clarify how our contribution is distinct from the two works you referenced.
> > >
> > > ---
> > >
> > > ### Clarification on Related Work
> > > - [1] Fang et al. focus on scaling laws for training dense retrieval models, particularly analyzing how *model and data size affect retrieval accuracy.*
> > > - [2] Shao et al. demonstrate that a larger retrieval corpus can improve the performance of LMs on downstream tasks by providing more relevant context.
> > >
> > > Both are valuable contributions to the retrieval community. However, **our focus is fundamentally different, as we explore scaling laws for memory utilization** in GUI agents, a domain with unique constraints and opportunities.
> > >
> > > ### Our Distinct Contribution
> > >
> > > In our paper, we empirically validate two scaling dynamics unique to the GUI agent setting:
> > >
> > > - **Memory Bank Size Scaling**: Increasing the size of the trajectory memory bank consistently improves agent performance.
> > >
> > > - **Inference-Time Sample Scaling with Continuous Memory**: Adding more memory samples during inference leads to better performance. *This is not feasible with discrete memory due to context length constraints, but is made possible by our Continuous Memory (CoMEM) mechanism.*
> > >
> > > To support and operationalize these findings, we also develop a fully automated data flywheel and plan to open-source our 100k trajectory dataset to benefit the broader research community — contingent on acceptance.
> > >
> > > ---
> > >
> > > We hope this clarification helps emphasize the novelty, scope, and impact of our work within the agent memory and GUI interaction context. Thank you again for your engagement and for considering a score adjustment!
> > >
> > > Best regards,
> > > The authors of *Auto-scaling Continuous Memory for GUI Agent*

---

### Official Review · Reviewer_Yviu · 2025-11-01

**Soundness:** 3
**Presentation:** 3
**Contribution:** 3
**Rating:** 6
**Confidence:** 3

**Summary:**

The paper introduces continuous memory, where each GUI trajectory is compressed by the VLM’s own encoder into a fixed-length sequence of continuous embeddings. During inference, retrieved memory vectors are directly injected into the VLM’s input embedding layer, avoiding context explosion while preserving visual detail. The authors also propose an automatic data pipeline that automatically constructs the memory database using open-source VLMs and search engines. They report a total cost of $553 for collecting 15,145 trajectories across 6,676 environments. The approach is evaluated on MMInA, Multimodal-Mind2Web, and WebVoyager benchmarks against text-based memory and various baselines. Results show consistent performance gains as memory scale and retrieval depth increase, while text memory degrades with longer prompts.

**Strengths:**

- The work offers a pragmatic solution by injecting external experience as continuous vectors directly into the VLM, circumventing long-context limitations while retaining fine-grained GUI visual information. Combined with a low-cost automated data flywheel, it demonstrates that open-source 7B-scale models can rival or surpass closed-source systems.
- Memory vectors are integrated at the embedding layer rather than concatenated as text, resulting in a cleaner architecture and predictable inference cost. The use of Q-Former + LoRA (only 1.2% of parameters) also facilitates efficient adaptation.
- The data collection process is scalable.
- The authors provide systematic evidence that performance scales monotonically with memory size and retrieval depth, while text-based memory deteriorates under longer contexts.

**Weaknesses:**

- Incomplete memory write/forget policy: The work focuses on the read path but lacks a quantitative analysis for when and how to write, deduplicate, or prune memories.
- The text-memory baseline is highly implementation-sensitive (e.g., summarization, structure, retrieval prompting). Insufficient disclosure of its setup may exaggerate the gap with continuous memory.
- Retrieval quality and error propagation: CLIP-style keys with FAISS nearest-neighbor retrieval may fail under small UI changes or noisy web elements (ads/pop-ups). If noisy memories are blindly injected, they may introduce strong interference. No robustness or gating analysis is presented.

**Questions:**

- Can the authors provide an evaluation using an open source VLM for the data flywheel?
- What motivated the choice of eight embeddings per trajectory? How does performance trade off with 4/16/32?

---

> ### Author Response · Authors · 2025-11-20
> **Response to Reviewer Yviu: part1**
>
> Thanks for your thoughtful comments! We appreciate the recognition of the challenges in memory systems and value the critical insights provided regarding memory management, baseline design, and retrieval robustness. Below, we respond to the identified weaknesses and questions, and provide clarifications and additional analysis to address the concerns raised.
>
> ### **Response to Weakness 1: Incomplete memory write/deduplicate/prune policy**
>
> We appreciate the reviewer’s concern regarding the completeness of our memory management policy. Below, we clarify how memory write, deduplication, and pruning are handled in our system.
>
> **Memory Write:**
> Our memory design is integrated and naturally supports memory updates. During implementation, we first collect a large number of diverse trajectories using our automated data flywheel and write them into the memory bank. At inference time, each successful trajectory is saved and written back into the memory bank so it can be reused in future tasks.
>
> **Memory Deduplication:**
> When constructing the memory bank, we ensure diversity across four dimensions to reduce redundancy:
>
> - *Domain Diversity* – We cover 16 domains including shopping, academic, news, government, and travel.
> - *Environment Diversity* – We gathered 6,676 distinct websites, each offering different layouts and navigation paths.
> - *Task Diversity* – For each website, we used a VLM to generate 10 varied questions, encouraging diverse user interactions.
> - *Trajectory Diversity* – Even for similar tasks, the agent may follow different UI paths, leading to unique trajectories.
>
> Figure 1 in our paper shows that more diverse memories are beneficial; hence, each retained memory is considered to have value and is preserved.
>
> **Memory Pruning and Quality Control:**
> While we do not perform online memory eviction during inference, we apply offline pruning and deduplication during memory construction to ensure high quality:
>
> - Only successful trajectories are retained.
> - Rule-based heuristics remove redundant or erroneous steps (e.g., repeated clicks, ineffective scrolling).
> - Domain-aware filtering is applied during retrieval to ensure relevance.
>
> These combined steps help maintain memory quality and prevent low-value or noisy entries from polluting the memory bank.

---

> ### Author Response · Authors · 2025-11-20
> **Response to Reviewer Yviu: part2**
>
> ### **Response to Weakness 2: Text-based Memory Baseline**
>
> In our main experiments, we implemented a text-based memory baseline using the exact same retrieval pipeline as CoMEM, and the retrieval details can be found in response to Weakness 1. After retrieval, each retrieved trajectory was converted into a textual format by:
> - Concatenating the original task instruction with the step-by-step actions taken during the trajectory.
> - Adding these text entries directly into the prompt as in-context examples.
>
> Thus, *the only difference between the text-based and continuous memory variants lies in the injection mechanism and modality*:
>
> - Text-based memory is inserted as raw tokens into the prompt. Screenshots were not added because of the context window length constraint.
> - CoMEM, by contrast, compresses trajectories into fixed-length embeddings and prepends them to the input layer, preserving multimodal information.
>
> To further address your concern, we implemented two advanced text-memory baselines inspired by recent literature:
>
> - *Agent Workflow Memory (AWM)* [1]: Induces reusable workflow templates from successful trajectories and injects them into the prompt.
> - *Reasoning Bank* [2]: Extracts memory from both successful and failed trajectories, allowing the agent to learn from both experience and reflection.
>
> These methods represent stronger and more structured approaches to text-based memory. All implementations use the same retrieval setup as CoMEM and are injected into the prompt using carefully designed templates. As shown in the result table, these structured memory approaches provide modest improvements in some domains but remain inconsistent and significantly underperform CoMEM. This clearly shows two key points:
>
> - Text-based memory is indeed more brittle and variable due to reliance on prompt formatting and token limits.
> - CoMEM offers robust and scalable gains by leveraging compact, multimodal embeddings that generalize better across domains.
>
> **Table: Performance Comparison Across Domains**
>
> | Method                     | MMInA-Wikipedia | Mind2Web-Travel | Mind2Web-Info  | Mind2Web-Service | Webvoyager |
> |---------------------------|-----------|--------|-------|---------|------------|
> | Qwen2.5-VL                | 36.7%     | 9.5%   | 9.6%  | 17.3%   | 40.0%      |
> | + Text-based Memory       | 34.2%     | 17.8%  | 12.7% | 16.6%   | 44.0%      |
> | + Agent-Workflow-Memory     | 37.7%    | 12.5%  | 14.4% | 17.2%   | 44.0%     |
> | + Reasoning Bank            | 37.3%    | 12.0%  | 15.5% | 16.4%   | 41.2%     |
> | + CoMEM                   | **47.4%**     | **18.8%**  | **26.5%** | **17.7%**   | **54.5%**      |
>
> [1] Wang, Zhiruo, et al. “Agent Workflow Memory.” arXiv preprint arXiv:2409.07429 (2024).
>
> [2] Ouyang, Siru, et al. “ReasoningBank: Scaling Agent Self-Evolving with Reasoning Memory.” arXiv preprint arXiv:2509.25140 (2025).

---

> ### Author Response · Authors · 2025-11-20
> **Response to Reviewer Yviu: part3**
>
> ### **Response to Weakness 3: CLIP+FAISS Coarse Retrieval and Error Propagation**
>
> We agree that retrieval quality is critical in memory-augmented systems and have taken several steps to mitigate these risks.
>
> In our current implementation, CLIP+FAISS is used as the first-stage retriever. To enhance robustness and filter out irrelevant or noisy candidates, we apply additional layers of filtering:
>
> - *Multimodal Retrieval*: We compute similarity using both image and text between the current task and each candidate memory (represented by initial screenshot + query).
> - *Domain-Aware Filtering*: Tasks are tagged with domain labels (e.g., shopping, travel), and we prioritize memory from the same domain to reduce semantic drift.
> - *Trajectory Quality Control*: Only successful trajectories are included in the memory bank. During memory construction, we filter out redundant or noisy UI actions using rule-based heuristics.
>
> These measures already provide reasonable robustness, as supported by CoMEM’s strong performance in our main results.
>
> To directly address the reviewer’s point about robustness under noisy memory injection and the lack of gating analysis, we performed two targeted evaluations:
>
> - *VLM-based Gating*: We added a lightweight second-stage filter using a vision-language model (VLM). After retrieving the top-30 candidates with CLIP+FAISS, we use the VLM to assess each trajectory’s relevance to the current task and retain the top-10 for memory encoding. This improves retrieval precision without modifying the memory architecture.
>
> - *Noisy Memory Stress Test*: We injected low-quality, irrelevant memory trajectories into the retrieval pool. While performance dropped compared to clean memory, CoMEM still consistently outperformed the base agent, demonstrating the robustness of our continuous memory design.
>
> **Retrieval Robustness: Quality Gating and Stress Test Results**
>
> | Method                     | MMInA-Wikipedia | Mind2Web-Travel | Mind2Web-Info  | Mind2Web-Service | Webvoyager |
> |--------------------------|-----------|--------|-------|---------|------------|
> | Qwen2.5-VL                | 36.7%     | 9.5%   | 9.6%  | 17.3%   | 40.0%      |
> | + CoMEM                  | 47.4%     | 18.8%  | 26.5% | 17.7%   | 54.5%      |
> | + VLM gating + CoMEM     | 48.3%     | 19.6%  | 28.6% | 19.6%   | 56.5%      |
> | + Noisy Memory + CoMEM   | 44.7%     | 17.1%  | 21.7% | 16.7%   | 49.0%      |
>
> These results confirm that retrieval quality has a meaningful impact, and CoMEM is compatible with more sophisticated reranking strategies (e.g., cross-encoders or hybrid retrievers). We view this compatibility and robustness as key strengths of our modular design.

---

> ### Author Response · Authors · 2025-11-20
> **Response to Reviewer Yviu: part4**
>
> ### **Response to Question 1: Evaluation with an Open-Source VLM for the Data Flywheel**
>
> We recognize the importance of evaluating the quality of data generated by our flywheel. In our current setup, we have already taken steps to ensure memory quality and also provide an evaluation using both automated models and human oversight:
>
> - *Open-Source VLM Validator*: For trajectory validation, we use SEAgent-1.0-7B [1], a fine-tuned open-source vision-language model specifically trained for GUI task verification.
>
> - *Rule-Based Filtering*: During memory construction, we apply deterministic heuristics to eliminate failed or redundant trajectories. Only successful and unique sequences are retained in the final memory bank.
>
> - *Evaluation with Human and LLM Raters*: To benchmark the quality of the stored trajectories, we performed a human-in-the-loop evaluation. We randomly sampled 40 trajectories and rated each three times using both GPT-4o and human annotators. Results show high alignment, we have attached sampled trajectories and evaluation records in the `trajectory_evaluation` folder of the supplementary file:
>   - GPT-4o: *90.83%* accuracy
>   - Human raters: *92.50%* accuracy
>
> These results indicate that the memory bank is of high fidelity, and that our open-source model is sufficiently reliable for use in the data flywheel. We agree that as the system scales, further automation and freshness checks will be important, and we are actively exploring additional open-source VLMs for future iterations. All trajectories and evaluation records are available in the supplementary materials.
>
> ---
>
> ### **Response to Question 2: Why use 8 token embeddings**
>
> Our goal is to compress each memory trajectory as much as possible while preserving key decision-relevant information. This enables the agent to scale up the number of memories it can attend to, as shown in Figure 1 of the paper. The number of tokens per trajectory is a tunable hyperparameter, and we found 8 to be the best trade-off in practice. To support this, we conducted ablation experiments on the MMInA shopping domain, varying the number of retrieved memory samples using memory encoders trained with 4, 8, or 16 tokens per trajectory. Due to resource constraints, we were unable to include the 32-token variant.
>
> - *4 tokens*: Too compressed to retain sufficient task-relevant information. Performance improves initially but saturates quickly.
> - *16 tokens*: Performs well with few memories, but degrades as more are added due to input length saturation. This is likely because the memory encoder was trained with only 3 samples, so it doesn’t generalize well to 30+ samples. We believe this is a result of training–inference mismatch caused by compute constraints. With more training samples, the 16-token variant could likely also exhibit scaling behavior.
> - *8 tokens*: Achieves *strong performance while maintaining scalability*, allowing a larger number of memory items to be injected without hurting the model’s capacity.
>
> Thus, *8 tokens strikes the best balance* between *representation quality* and *scalability* for continuous memory injection.
>
> | Memory Sample Size                       | 0     | 3     | 10    | 20    | 30    | 50    | 70    | 90    | 100   |
> |-----------------------------|-------|-------|-------|-------|-------|-------|-------|-------|--------|
> | Text-Memory                 | 15.5% | 22.0% | 30.5% | 31.0%   | 29.0% | 25.0% | 21.0%   | 22.0%   | 21.5%  |
> | CoMEM (4 token)    | 15.5% | 30.5% | 32.0% | 35.0% | 37.0% | 38.5% | 37.5% | 39.0% | 40.0%  |
> | CoMEM (8 token)    | 15.5% | 41.5% | 45.0% | 43.0%   | 42.0% | 43.5% | 44.5% | 45.5% | 46.0%  |
> | CoMEM (16 token)   | 15.5% | 42.5% | 46.5% | 47.0% | 44.0% | 41.5% | 38.5% | 39.0% | 38.0%  |
>
> [1] Sun, Zeyi, et al. “SEAgent: Self-Evolving Computer Use Agent with Autonomous Learning from Experience.” arXiv preprint arXiv:2508.04700 (2025).

---

> ### Author Response · Authors · 2025-11-27
> **Follow Up on Rebuttal**
>
> Dear Reviewer Yviu,
>
> Thank you for your time and constructive feedback! We’re writing to kindly follow up and see if you’ve had a chance to review our rebuttal. We’ve responded your comments with additional experiments and clarifications, and we hope our response adequately resolves your concerns.
>
> So far, two reviewers have kindly acknowledged the improvements and indicated they will raise their scores. If the rebuttal resolves your questions, we would be sincerely grateful if you could consider updating your score. Your feedback is very important to us.
>
> We truly appreciate your input and look forward to your response!
>
> Best regards,
>
> The authors of *Auto-scaling Continuous Memory for GUI Agent*

---

### Official Review · Reviewer_Q6me · 2025-11-01

**Soundness:** 3
**Presentation:** 3
**Contribution:** 2
**Rating:** 4
**Confidence:** 4

**Summary:**

The paper proposes a plug-and-play “continuous memory” for GUI agents. Past multi-modal trajectories (screens + actions) are encoded by a small Q-Former into a handful of fixed-length vectors and prepended as a soft prefix at the input-embedding layer, avoiding prompt bloat while letting the backbone attend to relevant experience. At runtime, a CLIP+FAISS retriever pulls top-k similar trajectories, the memory encoder produces the vectors, and the VLM consumes them with the current screen/instruction. A low-cost flywheel keeps expanding the memory bank. Experiments show strong gains on real-world web benchmarks.

**Strengths:**

Clean, architecture-agnostic injection, lightweight tuning and low inference overhead make it practical. The approach exhibits a clear scaling trend, more/better memories and deeper top-k retrieval steadily help, while delivering competitive results on real web GUIs and lifting specialized baselines too. The automated data flywheel keeps costs down and coverage growing, and the design plays nicely with existing agent stacks (retrieval, planning, tool use), making it easy to slot into production-style GUI agents

**Weaknesses:**

Q1: Gains on OSWorld are modest, suggesting the memory mostly captures web-navigation regularities and doesn’t transfer cleanly to desktop/app workflows. If the base model switches (e.g., to UI-TARS) and improvements remain small, that points to a real domain gap rather than an underpowered baseline.

Q2: The CLIP+FAISS first-stage retrieval is coarse, and the injected memory prefix is cheap for the model to trust. When retrieval is slightly off, those few vectors can still steer attention the wrong way. The paper lacks stress tests for retrieval noise and stronger reranking/quality gates (e.g., cross-encoder re-rank, confidence gating, or learned trust of memory).


Q3: Collapsing long, multi-step GUI traces into ~8 learned vectors is aggressive. It likely drops temporal dependencies and subtle causal cues (order of actions, transient states), especially under large layout shifts. The scaling curves are encouraging, but there isn’t a convincing analysis that these embeddings retain the decision-critical bits in hard OOD cases versus capturing surface similarity.

Q4: The low-cost loop (discover → synthesize → rollout → auto-verify) is efficient but invites bias and label noise. Without clear validator error rates, duplicate/failure audits, or freshness governance, the memory bank can accumulate skewed or stale experiences. Ethical/robustness issues (privacy, site policies) are acknowledged but not operationalized.

**Questions:**

see above weakness

---

> ### Author Response · Authors · 2025-11-20
> **Response to Reviewer Q6me: part 1**
>
> ## Response to Reviewer Q6me
>
> We thank the reviewer for the detailed and thoughtful feedback. Below, we address each of the raised concerns individually. We provide additional experiments, clarifications, and analysis to support the robustness and generalizability of our framework. We hope our responses clarify the contributions and empirical insights of the work.
>
> ---
>
> ### **Response to Weakness 1: Gains on OSWorld and Domain Transfer**
>
> We acknowledge that the effectiveness of retrieved memory is influenced by *domain alignment*. To probe this point, we conducted additional experiments using *domain-aligned memory*. Specifically, we used *13,750 trajectories* from the training set of [GUI-360°](https://arxiv.org/abs/2511.04307) [1] as our memory bank, a dataset covering desktop applications like Microsoft Word, Excel, and PowerPoint.
>
> When memory is drawn from this desktop/app domain, we observe *substantially higher gains* on both *GUI-Oddsey* and *OSWorld*. These results are summarized in the table below and demonstrate that our framework generalizes well across domains—*especially when memory is matched appropriately*.
>
> In addition, as shown in Table 3 of our paper, experiments demonstrate that even when memory is collected from a completely different domain (web navigation) and applied to desktop/app environments (e.g., Word, PowerPoint, VS Code), our continuous memory method (CoMEM) still improves performance over the baseline—achieving an *average absolute gain of +1.6%*. This suggests that CoMEM captures *abstract, transferable patterns* of GUI interaction (e.g., multi-step planning, grounding UI elements, feedback loops) that are *not specific to any single domain*.
>
> Moreover, our *auto-scaling data flywheel* plays a pivotal role in bridging domain gaps. Unlike prior work that relies on static, human-annotated datasets, our framework supports the *continuous collection of high-quality trajectories* from arbitrary domains—including desktop applications. This flexibility allows our agent to adapt quickly to novel environments and build *general-purpose memory banks* that span diverse UI paradigms.
>
>
> #### Cross-Domain Memory Transfer Results
>
> | Memory Type               | GUI-Oddsey (AMS) High-Level | GUI-Oddsey (AMS) Low-Level | OSWorld (SR) Office | OSWorld (SR) Daily | OSWorld (SR) Professional | OSWorld (SR) Overall | Avg (%) |
> |---------------------------|-----------------------------|-----------------------------|---------------------|--------------------|----------------------------|----------------------|---------|
> | **Baseline**              | 22.4%                       | 45.6%                       | 24.7%               | 25.6%              | 60.2%                      | 26.4%                | 35.7    |
> | Web-domain + Text Memory  | 24.4%                       | 37.4%                       | 23.1%               | 23.1%              | 57.1%                      | 24.7%                | 33.0    |
> | Web-domain + CoMEM        | 27.4%                       | 44.9%                       | 25.1%               | 28.2%              | 60.9%                      | 26.7%                | 37.3    |
> | Desktop-domain + Text Mem | 26.9%                       | 48.8%                       | 27.3%               | 28.2%              | 63.3%                      | 27.8%                | 38.9    |
> | **Desktop-domain + CoMEM**| **31.0%**                   | **52.2%**                   | **30.7%**           | **30.8%**          | **67.4%**                  | **30.0%**            | **42.4** |
>
> [1] Liu, Zihan, et al. “GUI-360°: A Benchmark for Multi-Platform GUI Agents.” arXiv preprint arXiv:2511.04307 (2025).

---

> ### Author Response · Authors · 2025-11-20
> **Response to Reviewer Q6me: part2**
>
> ### **Response to Weakness 2: CLIP+FAISS coarse retrieval issue**
>
> Retrieval quality plays a critical role in memory-augmented modeling, and our approach is compatible with other retrieval techniques.
>
> In this paper, we use CLIP+FAISS as the first-stage retriever, then apply several additional filters to ensure retrieval quality:
>
> - *Multimodal Retrieval*: We compute similarity between the current task (image + text) and the initial screenshot + query from each candidate trajectory.
> - *Domain-Aware Filtering*: Each task is tagged with a domain label (e.g., shopping, travel, info), and we prioritize memory items from the same domain to reduce semantic drift.
> - *Trajectory-Level Quality Control*: Only successful trajectories are stored in memory, and we prune noisy or redundant steps using rule-based heuristics during memory construction.
>
> These steps already provide a reasonable level of robustness, as evidenced by the strong performance of CoMEM reported in our main results.
>
> Besides, our continuous memory design accepts any retrieved trajectory and compresses it into a fixed-length embedding block. It means that more sophisticated retrieval pipelines (e.g., cross-encoder re-ranking) can be plugged in without modifying the memory structure or the agent architecture. We view this flexibility as one of CoMEM’s key strengths.
>
> To directly address the reviewer’s suggestion, we implemented a VLM-based quality gating mechanism as a lightweight second-stage filter. Specifically, we:
>
> - Retrieve the top-30 candidates using CLIP+FAISS.
> - Use a VLM to judge each candidate trajectory’s relevance and quality for the current task.
> - Keep only the top-10 filtered items to encode into memory embeddings.
>
> As shown in the table below, this simple gating mechanism leads to consistent performance improvements across domains. This confirms that retrieval quality is indeed important, and that CoMEM benefits from stronger reranking — a promising direction for future work.
>
> Notably, we also performed a *stress test* by introducing noisy memory trajectories into the retrieval pool. As shown in the last row, while performance drops slightly compared to clean memory, *CoMEM still maintains significant gains over the base agent*, highlighting the robustness of our continuous memory design.
>
> #### CoMEM Stress Test and Quality Gating Results
>
> | Model Variant             | Wikipedia | Travel | Info  | Service | Webvoyager |
> |--------------------------|-----------|--------|-------|---------|------------|
> | Qwen2.5-VL                | 36.7%     | 9.5%   | 9.6%  | 17.3%   | 40.0%      |
> | + CoMEM                  | 47.4%     | 18.8%  | 26.5% | 17.7%   | 54.5%      |
> | + VLM gating + CoMEM     | 48.3%     | 19.6%  | 28.6% | 19.6%   | 56.5%      |
> | + Noisy Memory + CoMEM   | 44.7%     | 17.1%  | 21.7% | 16.7%   | 49.0%      |

---

> ### Author Response · Authors · 2025-11-20
> **Response to Reviewer Q6me: part 3**
>
> ### **Response to Weakness 3: Compression ≠ Lossless Copy — It Is a Learned Distillation**
>
> Our goal is *not* to preserve every low-level detail from the trajectory, but to distill abstract, high-utility knowledge that aids future reasoning and planning. The memory encoder — composed of a VLM + Q-former — is trained to extract the most salient planning cues from raw trajectories. This learned abstraction enables generalization across layout shifts and domain variations.
>
> To empirically validate that compressed memory embeddings encode decision-critical information, we conducted a *rule extraction experiment*:
>
> - We randomly sampled 50 tasks from the WebVoyager dataset.
> - For each, we retrieved top-10 trajectories using CLIP.
> - We compared two conditions:
>   - (a) The agent observes only the compressed memory embeddings (CoMEM).
>   - (b) The agent receives the raw text and screenshot trajectory in the prompt.
>
> In both cases, the model is asked to summarize useful rules that can guide future task solving.
>
> **Findings:**
>
> - The CoMEM condition consistently produced *abstract, generalizable strategies*
>   (e.g., *"check the top bar for filters"*, *"scroll until you see icons"*).
> - The text-prompt condition often produced *literal summaries* that overfit to specific UI layouts or labels, occasionally leading to *misleading heuristics* under domain shifts.
>
> This suggests that CoMEM doesn't just encode surface similarity — it learns task-relevant *invariants* that are robust under OOD perturbations. (Also supports Response to Weakness 1.)
>
> ---
>
> #### Example Comparison
>
> Below is an illustrative example comparing responses from the two conditions.
> (Full results across 50 tasks are available in the `memory_check` folder of supplementary file.)
>
> ---
>
> **Task ID**: `food_tasks_food_V1_new_428`
> **CoMEM Response**:
> ```
> ### Key Insights for Finding Holiday-Themed Recipes
>
> #### Decision Rules:
> - Begin by searching for relevant keywords.
> - Navigate to the Recipes section within the Holidays category.
> - Use filters to narrow down to holiday-themed recipes.
> - Select a suitable recipe based on visibility and relevance.
>
> #### Success Factors:
> - Effective use of search functionality.
> - Efficient navigation through categories and subcategories.
> - Utilization of filtering options to refine results.
> - Quick selection of a relevant recipe.
>
> #### Common Mistakes:
> - Ineffective searching with generic terms.
> - Overlooking or using incorrect filters.
> - Delaying decisions or exploring unrelated sections.
> - Impulsive selection of recipes without verification.
>
> These insights can help users efficiently locate and read holiday-themed recipes.
> ```
>
>
> **Text Memory Response**:
> ```
> ### High-Level Behavioral Patterns and Rules Extraction
>
> #### Decision Rules
> 1. Navigate Directly to Target Section: The user consistently navigates directly to the "Holidays" section by clicking on its link, indicating a clear goal orientation towards finding holiday-themed recipes.
> 2. Use Visible Links: The user relies on visible links within the interface to achieve their goal, suggesting a preference for straightforward navigation paths.
>
> #### Success Factors
> 1. Immediate Access to Content: Successfully clicking on the "Holidays" link leads directly to the relevant section, ensuring quick access to holiday-themed content.
> 2. Clear Navigation Path: The presence of a dedicated "Holidays" link at the bottom of the page facilitates easy navigation, making the task efficient.
>
> #### Common Mistakes
> 1. Overlooking Hidden Links: There were no instances of overlooking hidden or less obvious links, but it’s important to note that if there were, it would have been a mistake to not use visible links.
> 2. Inconsistent Clicking: The user clicked on the same element multiple times, which might indicate hesitation or uncertainty, though this was resolved successfully without issues.
> ```

---

> ### Author Response · Authors · 2025-11-20
> **Response to Reviewer Q6me: part 4**
>
> ### **Response to Weakness 4: Memory Quality Control and Ethical Safeguards**
>
> We consider both memory quality control and ethical safeguards in our current design.
>
>  ---
> ### Memory Quality Control
> Below are our current mechanisms:
>
> - *LLM-as-Judge*: We use a fine-tuned validation model (SEAgent-1.0-7B [1]) to evaluate trajectory success. This model is trained specifically for GUI task verification.
>
> - *Rule-Based Filtering*: We additionally apply heuristics to remove failed and redundant trajectories during memory construction. Only *successful* and *non-duplicated* steps are retained.
>
> - *Human + LLM Evaluation*: To assess memory quality, we conducted a manual audit:
>   We randomly sampled 40 trajectories and evaluated each 3 times using both *GPT-4o* and human raters.
>   Final accuracy is above 90%, with:
>   - GPT-4o — *90.83%*
>   - Human — *92.50%*
>
> These results suggest that our memory bank maintains high quality and low noise. Though we acknowledge that scaling up may require more automated auditing and freshness checks, we provide these trajectories and evaluation records in the `trajectory_evaluation` folder in the supplementary file.
>
> ---
>
> ### Ethical and Robustness Safeguards
>
> We have designed *three layers of ethical control* throughout the data pipeline:
>
> - *Website Collection*:  We use *SerpAPI* [2], which offers legal protections and only indexes publicly accessible content.
>   Seed queries come from public datasets, ensuring relevance and safety.
>
> - *Task Generation*:  Unsafe or sensitive tasks are filtered using an LLM to avoid generating inappropriate or harmful interactions.
>
> - *Task Execution*:  If a website blocks access or detects automation, the agent immediately halts execution to respect site policies and avoid scraping protected content.
>
> ---
>
> We are committed to maintaining ethical standards and will explore adding *freshness tracking*, *deduplication audits*, and *privacy filters* in future iterations.
>
> [1] Sun, Zeyi, et al. “SEAgent: Self-Evolving Computer Use Agent with Autonomous Learning from Experience.” arXiv preprint arXiv:2508.04700 (2025).
>
> [2] SerpAPI. “Google Search API.” SerpAPI, 2023. https://serpapi.com

---

> ### Author Response · Authors · 2025-11-27
> **Follow Up on Rebuttal**
>
> Dear Reviewer Q6me,
>
> Thank you again for your time and thoughtful feedback! We wanted to kindly follow up to see if you had a chance to review our rebuttal. We’ve made substantial updates and conducted additional experiments based on your comments, and we hope they address your concerns.
>
> So far, two reviewers have kindly acknowledged the improvements and indicated they will raise their scores.  If the rebuttal resolves your questions, we would be sincerely grateful if you could consider updating your score. Your feedback is very important to us.
>
> Thanks again for your comments, and looking forward to your update!
>
> Best Regards,
>
> The authors of *Auto-scaling Continuous Memory for GUI Agent*

---

### Author Response · Authors · 2025-11-26
**Reminder and Summary of Rebuttal**

Dear Reviewers,

We sincerely thank you for your valuable time, thoughtful feedback, and constructive engagement during the review process. Your insights have been instrumental in shaping our submission, and we deeply appreciate the clarity and depth of your comments.

In response to your suggestions, we have conducted a broad set of additional experiments and provided detailed clarifications to strengthen our work. However, we have not yet received responses from reviewers **Q6me**, **Yviu**, **BtWp**, and **31Nz**. Your feedback on our rebuttal would be greatly appreciated, as we are eager to know whether our efforts adequately address your questions and concerns.

---

We summarize our rebuttal contents in the following:

### Additional Experiments
- Memory Quality Control
    - **Experiment 1: Memory Bank Evaluation with Human + LLM Evaluation**: We randomly sampled 40 trajectories and evaluated each 3 times using both GPT-4o and human raters, achieving over 90% agreement on correctness.
    - **Experiment 2: Stress Test with Noisy Memory Retrieval**: We introduced 50% noise into retrieved memory samples and observed that CoMEM remains robust, maintaining strong performance under degraded memory quality.

- Memory Robustness
    - **Experiment 3: Comparison with advanced text-based memory baselines**: We added comparisons against Agent Workflow Memory (AWM) and Reasoning Bank, demonstrating that CoMEM outperforms both by a significant margin.
	- **Experiment 4: Effectiveness on Smaller Model**: CoMEM was applied to a 3B model and showed clear gains, indicating that our memory encoder effectively compensates for limited model capacity.
    - **Experiment 5: Token Embedding Count Ablation (4/8/16)**: We tested different token lengths (4/8/16) for memory embeddings. All outperform the baseline; 8 was selected as the optimal tradeoff between quality and scalability.

- Memory Generalizability
    - **Experiment 6: Cross-Domain Generalization on OSWorld + GUI-Odyssey**: We demonstrate CoMEM's effectiveness in desktop/app environments (e.g., Word, PowerPoint, VS Code) with both domain-aligned and not aligned memory — showing consistent improvements.
	-  **Experiment 7: VLM-Based Quality Gating for Retrieval Enhancement**: We added a second-stage quality gating mechanism using a VLM to filter retrieved memory samples, leading to further performance gains with minimal overhead.

### Clarifications and Improvements
- **Clarification 1: Compression ≠ Lossless Copy — It Is a Learned Distillation**: We clarified that CoMEM compresses by distilling abstract, generalizable knowledge — not by memorizing low-level details — as supported by our rule extraction experiments.
- **Clarification 2: Distinct Contributions of CoMEM**: CoMEM is more than a compression method — it is a scalable, general-purpose memory solution for long-horizon, cross-domain GUI agent tasks. Using trajectory-level CoMEM architecture, we validate Empirical scaling laws for memory size and retrieval depth, and develop an auto memory flywheel to provide data support for it.
- **Clarification 3: Modular and Easily Integratable with Other Methods**: Our VLM-based encoder is highly modular and integrates seamlessly with other retrieval-based methods; even simple CLIP-based retrieval achieves strong performance.
- **Clarification 4: Detailed Retrieval Criteria and Case Studies**: We provided a clearer explanation of how retrieval and memory construction are conducted, along with case studies to illustrate the process.
- **Clarification 5: Ethical and Robustness Safeguards**: To ensure responsible deployment, we implemented a three-layer ethical control pipeline (Website Collection → Task Generation → Task Execution), with plans for further safeguards including freshness tracking and privacy filtering.

---

We hope these additions clearly demonstrate the effectiveness, generality, and reliability of our proposed method. We are committed to incorporating these new results into the final version of the paper and further polishing the writing for clarity and precision after receiving final feedback.

Once again, we sincerely appreciate your reviews and kindly encourage any remaining reviewers to share their thoughts. Your input is invaluable to us.

Best regards,
The Authors of *Auto-scaling Continuous Memory for GUI Agent*

---

### Comment · Area_Chair_ejeZ · 2025-11-27
**Reviewers: please read the rebuttals**

Dear Reviewers,

Please read the authors' rebuttals and make necessary edits to your reviews.

Best,

AC

---

### Meta-Review · Area_Chair_9Gvi · 2026-01-01

**Summary:**

This submission proposes CoMEM, a “continuous memory” mechanism for GUI agents: past multimodal trajectories are compressed into a small fixed number of continuous embeddings (e.g., 8) and injected at the backbone’s input-embedding layer to avoid long-context prompt bloat while retaining multimodal information. The paper also introduces an automated data flywheel (discover → synthesize → rollout → verify) to expand the memory bank at low cost, and reports scaling trends with memory bank size and retrieval depth.

Across reviews, there is broad agreement that the system is practically motivated and that the embedding-level memory injection is a clean design choice. However, the reviews also converge on several decision-critical concerns: (i) novelty/positioning relative to existing memory and retrieval-based agent methods, (ii) robustness and failure modes when retrieval is wrong or memory is noisy (including potential steering), (iii) quality control and governance of the self-generated flywheel data (validator error, bias, duplicates, freshness, and ethics), and (iv) reproducibility clarity (retrieval configuration, curation criteria, and consistency of reported figures).

The rebuttal provides substantial additional experiments and clarifications (e.g., noisy-memory stress tests, VLM-based gating, stronger text-memory baselines, token-count ablations, small-model results, and limited audits). Several reviewers explicitly acknowledged that many concerns were addressed and indicated score increases. Nevertheless, the discussion still reflects meaningful disagreement about (a) whether the contribution is sufficiently distinct beyond combining/engineering known ingredients for this domain, and (b) whether the robustness/governance envelope is characterized convincingly enough for acceptance.

**Reviewer Concerns:**

### Addressed by the rebuttal (to a meaningful extent)

- Baseline strength: Authors added comparisons against more structured text-memory baselines (e.g., AWM, Reasoning Bank) rather than only a naive text memory, which directly responds to concerns about baseline weakness.

- Retrieval robustness (partial): Authors added a noisy-memory stress test (injecting noise into retrieved memories) and a second-stage VLM gating mechanism, both of which improve the evidence that the method can tolerate some retrieval imperfections.

- Token budget / compression choice: Authors provided token-count ablations (4/8/16) and justification for selecting 8 as a tradeoff.

- Small-model feasibility (partial): Authors added results on a 3B model showing improvements with CoMEM, addressing the request for evidence beyond 7B.

- Cross-domain transfer (partial): Authors provided additional transfer results using domain-aligned memory (and contrasted with mismatched memory), addressing domain-gap concerns at least empirically within the provided settings.

- Reproducibility details (partial): Authors expanded descriptions of retrieval (CLIP+FAISS), filtering heuristics, and cost accounting in responses, which helps—though see remaining issues below.

- Some reviewer concerns explicitly resolved: Reviewer BtWp stated that most concerns were addressed and indicated an increase; Reviewer 31Nz also updated their score and leaned accept after reviewing the rebuttal and added experiments.

### Still outstanding / not convincingly resolved (based on the discussion record)

- Contribution / novelty remains contested: Even after rebuttal, at least one reviewer still questioned whether the key contribution is primarily a compression/injection engineering choice plus an automated collection loop, and noted that “scaling law” claims are narrow/specific in light of broader retrieval scaling-law literature. The rebuttal clarifies distinctions, but the novelty debate remains.

- Robustness envelope is not fully characterized: The new stress tests and gating are helpful, but they do not yet pin down when the memory harms performance (e.g., under substantial UI layout drift / domain mismatch / compounding retrieval errors over long horizons). The risk that a short continuous prefix can strongly steer the policy when retrieval is off is mitigated but not fully closed.

- Flywheel governance and validator reliability at scale: The rebuttal adds limited audits (e.g., small-sample human/LLM agreement) and describes safety filters, but key questions remain about validator error rates at scale, duplicate/staleness handling, freshness policies, and long-term bias accumulation in the memory bank.

- Terminology and framing (“auto-scaling”): One reviewer flagged that “auto-scaling” can be misleading; authors agreed to revise terminology. This is addressable editorially, but it reflects broader overclaim/positioning risk.

- Reproducibility still depends on paper integration: While responses clarify many details, it remains uncertain (from the discussion record) whether all rebuttal experiments/clarifications are cleanly integrated into the revised manuscript, and whether readers can reproduce the pipeline without relying on rebuttal-only details.

**Reviewer Scores:**

- Reviewer BtWp explicitly stated that most concerns were addressed and that they will adjust their score accordingly, while noting residual concerns about contribution/positioning.

- Reviewer 31Nz explicitly stated they updated their score after reading the rebuttal and are leaning toward acceptance, while still flagging challenges around generally applicable tuning/retrieval under frequent UI updates and asking that rebuttal additions be clearly reflected in the final manuscript.

- No post-rebuttal score update is recorded in the thread for Reviewer Q6me or Reviewer Yviu.

---

### Decision · Program_Chairs · 2026-01-26

Reject